# Planning in Markov Decision Processes with Gap-Dependent Sample Complexity

**Anders Jonsson**
Universitat Pompeu Fabra
anders.jonsson@upf.edu

**Emilie Kaufmann**
CNRS & ULille (CRIStAL), Inria Scool
emilie.kaufmann@univ-lille.fr

**Pierre Ménard**
Inria Lille, Scool team
pierre.menard@inria.fr

**Omar Darwiche Domingues**
Inria Lille, Scool team
omar.darwiche-domingues@inria.fr

**Edouard Leurent**
Renault & Inria Lille, Scool team
edouard.leurent@inria.fr

**Michal Valko**
DeepMind Paris
valkom@deepmind.com

## Abstract

We propose `MDP-GapE`, a new trajectory-based Monte-Carlo Tree Search algorithm for planning in a Markov Decision Process in which transitions have a finite support. We prove an upper bound on the number of calls to the generative model needed for `MDP-GapE` to identify a near-optimal action with high probability. This problem-dependent *sample complexity* result is expressed in terms of the *sub-optimality gaps* of the state-action pairs that are visited during exploration. Our experiments reveal that `MDP-GapE` is also effective in practice, in contrast with other algorithms with sample complexity guarantees in the fixed-confidence setting, that are mostly theoretical.

## 1 Introduction

In reinforcement learning (RL), an agent repeatedly takes *actions* and observes *rewards* in an unknown environment described by a *state*. Formally, the environment is a Markov Decision Process (MDP) $\mathcal{M} = \langle \mathcal{S}, \mathcal{A}, p, r \rangle$, where $\mathcal{S}$ is the state space, $\mathcal{A}$ the action space, $p = \{p_h\}_{h \geq 1}$ a set of transition kernels and $r = \{r_h\}_{h \geq 1}$ a set of reward functions. By taking action $a$ in state $s$ at step $h$, the agent reaches a state $s'$ with probability $p_h(s'|s, a)$ and receives a random reward with mean $r_h(s, a)$. A common goal is to learn a policy $\pi = (\pi_h)_{h \geq 1}$ that maximizes cumulative reward by taking action $\pi_h(s)$ in state $s$ at step $h$. If the agent has access to a generative model, it may *plan* before acting by generating additional samples in order to improve its estimate of the best action to take next.

In this work, we consider *Monte-Carlo planning* as the task of recommending a good action to be taken by the agent in a given state $s_1$, by using samples gathered from a generative model. Let $Q^\star(s_1, a)$ be the maximum cumulative reward, in expectation, that can be obtained from state $s_1$ by first taking action $a$, and let $\hat{a}_n$ be the recommended action after $n$ calls to the generative model. The quality of the action recommendation is measured by its *simple regret*, defined as $\bar{r}_n(\hat{a}_n) := V^\star(s_1) - Q^\star(s, \hat{a}_n)$, where $V^\star(s_1) := \max_a Q^\star(s_1, a)$.

We propose an algorithm in the *fixed confidence* setting $(\varepsilon, \delta)$: after $n$ calls to the generative model, the algorithm should return an action $\hat{a}_n$ such that $\bar{r}_n(\hat{a}_n) \leq \varepsilon$ with probability at least $1 - \delta$. We prove that its *sample complexity* $n$ is bounded in high probability by a quantity that depends on

Table 1: Different settings of planning algorithms in the literature

| Setting | Input | Output | Optimality criterion |
|---|---|---|---|
| (1) Fixed confidence (action-based) | $\varepsilon, \delta$ | $\widehat{a}_n$ | $\mathbb{P}\left(\bar{r}_n(\widehat{a}_n) \leq \varepsilon\right) \geq 1 - \delta$ |
| (2) Fixed confidence (value-based) | $\varepsilon, \delta$ | $\widehat{V}(s_1)$ | $\mathbb{P}\left(\left|\widehat{V}(s_1) - V^\star(s_1)\right| \leq \varepsilon\right) \geq 1 - \delta$ |
| (3) Fixed budget | $n$ (budget) | $\widehat{a}_n$ | $\mathbb{E}\left[\tilde{r}_n(\widehat{a}_n)\right]$ decreasing in $n$ |
| (4) Anytime | - | $\widehat{a}_n$ | $\mathbb{E}\left[\bar{r}_n(\widehat{a}_n)\right]$ decreasing in $n$ |

Table 2: Algorithms with sample complexity guarantees

| Algorithm | Setting | Sample complexity | Remarks |
|---|---|---|---|
| Sparse Sampling [19] | (1)-(2) | $H^5(BK)^H/\varepsilon^2$ or $\varepsilon^{-\left(2+\frac{\log(BK)}{\log(1/\gamma)}\right)}$ | proved in Lemma 1 |
| OLOP [2] | (3) | $\varepsilon^{-\max\left(2,\frac{\log \kappa}{\log(1/\gamma)}\right)}$ | open loop, $\kappa \in [1, K]$ |
| OP [3] | (4) | $\varepsilon^{-\frac{\log \kappa}{\log(1/\gamma)}}$ | known MDP, $\kappa \in [1, BK]$ |
| BRUE [8] | (4) | $H^4(BK)^H/\Delta^2$ | minimal gap $\Delta$ |
| StOP [28] | (1) | $\varepsilon^{-\left(2+\frac{\log \kappa}{\log(1/\gamma)}+o(1)\right)}$ | $\kappa \in [1, BK]$ |
| TrailBlazer [13] | (2) | $\varepsilon^{-\max\left(2,\frac{\log(B\kappa)}{\log(1/\gamma)}+o(1)\right)}$ | $\kappa \in [1, K]$ |
| SmoothCruiser [14] | (2) | $\varepsilon^{-4}$ | only regularized MDPs |
| MDP-GapE (ours) | (1) | $\sum_{a_1 \in \mathcal{A}} \frac{H^2(BK)^{H-1}B}{(\Delta_1(s_1,a_1) \vee \Delta \vee \varepsilon)^2}$ | see Corollary 1 |

the sub-optimality gaps of the actions that are applicable in state $s_1$. We also provide experiments showing its effectiveness. The only assumption that we make on the MDP is that the support of the transition probabilities $p_h(\cdot|s,a)$ should have cardinality bounded by $B < \infty$, for all $s$, $a$ and $h$.

Monte-Carlo Tree Search (MCTS) is a form of Monte-Carlo planning that uses a *forward model* to sample transitions from the current state, as opposed to a full generative model that can sample anywhere. Most MCTS algorithms sample *trajectories* from the current state [1], and are widely used in *deterministic* games such as Go. The AlphaZero algorithm [26] guides planning using value and policy estimates to generate trajectories that improve these estimates. The MuZero algorithm [25] combines MCTS with a model-based method which has proven useful for *stochastic* environments. Hence *efficient Monte-Carlo planning* may be instrumental for learning better policies. Despite their empirical success, little is known about the sample complexity of state-of-the-art MCTS algorithms.

**Related work**   The earliest MCTS algorithm with theoretical guarantees is Sparse Sampling [19], whose sample complexity is polynomial in $1/\varepsilon$ in the case $B < \infty$ (see Lemma 1). However, it is not trajectory-based and does not select actions adaptively, making it very inefficient in practice.

Since then, adaptive planning algorithms with small sample complexities have been proposed in different settings with different optimality criteria. In Table 1, we summarize the most common settings, and in Table 2, we show the sample complexity of related algorithms (omitting logarithmic terms and constants) when $B < \infty$. Algorithms are either designed for a discounted setting with $\gamma < 1$ or an episodic setting with horizon $H$. Sample complexities are stated in terms of the accuracy $\varepsilon$ (for algorithms with fixed-budget guarantees we solve $\mathbb{E}\left[\bar{r}_n\right] = \varepsilon$ for $n$), the number of actions $K$, the horizon $H$ or the discount factor $\gamma$ and a problem-dependent quantity $\kappa$ which is a notion of branching factor of near-optimal nodes whose exact definition varies.

A first category of algorithms rely on optimistic planning [23], and require additional assumptions: a deterministic MDP [15], the *open loop* setting [2, 21] in which policies are sequences of actions instead of state-action mappings (the two are equivalent in MDPs with deterministic transitions), or an MDP with known parameters [3]. For MDPs with stochastic and unknown transitions, polynomial sample complexities have been obtained for StOP [28], TrailBlazer [13] and SmoothCruiser [14] but the three algorithms suffer from numerical inefficiency, even for $B < \infty$. Indeed, StOP explicitly reasons about policies and storing them is very costly, while TrailBlazer and SmoothCruiser require a very large amount of recursive calls even for small MDPs. We remark that popular MCTS algorithms such as UCT [20] are not $(\varepsilon, \delta)$-correct and do not have provably small sample complexities.

In the setting $B < \infty$, BRUE [8] is a trajectory-based algorithm that is anytime and whose sample complexity depends on the smallest sub-optimality gap $\Delta := \min_{a \neq a^\star}\left(V^\star(s_1) - Q^\star(s_1, a)\right)$. For

planning in deterministic games, gap-dependent sample complexity bounds were previously provided in a fixed-confidence setting [16, 18]. Our proposal, `MDP-GapE`, can be viewed as a non-trivial adaptation of the `UGapE-MCTS` algorithm [18] to planning in MDPs. The defining property of `MDP-GapE` is that it uses a best arm identification algorithm, UGapE [10], to select the first action in a trajectory, and performs optimistic planning thereafter, which helps refining confidence intervals on the intermediate Q-values. Best arm identification tools have been previously used for planning in MDPs [24, 29] and UGapE also served as a building block for StOP [28].

Finally, going beyond worst-case guarantees for RL is an active research direction, and in a different context gap-dependent bounds on the regret have recently been established for tabular MDPs [27, 30].

**Contributions**    We present `MDP-GapE`, a new MCTS algorithm for planning in the setting $B < \infty$. `MDP-GapE` performs efficient Monte-Carlo planning in the following sense: First, it is a simple trajectory-based algorithm which performs well in practice and only relies on a forward model. Second, while most practical MCTS algorithms are not well understood theoretically, we prove upper bounds on the sample complexity of `MDP-GapE`. Our bounds depend on the *sub-optimality gaps* associated to the state-action pairs encountered during exploration. This is in contrast to StOP and TrailBlazer, two algorithms for the same setting, whose guarantees depend on a notion of near-optimal nodes which can be harder to interpret, and that can be inefficient in practice. In the anytime setting, BRUE also features a gap-dependent sample complexity, but only through the worst-case gap $\Delta$ defined above. As can be seen in Table 1, the upper bound for `MDP-GapE` given in Corollary 1 improves over that of BRUE as it features the gap of each possible first action $a_1$, $\Delta_1(s_1, a_1) = V^\star(s_1) - Q_1^\star(s_1, a_1)$, and scales better with the planning horizon $H$. Furthermore, our proof technique relates the *pseudo-counts* of any trajectory prefix to the gaps of state-action pairs on this trajectory, which evidences the fact that `MDP-GapE` does not explore trajectories uniformly.

## 2   Learning Framework and Notation

We consider a *discounted episodic setting* where $H \in \mathbb{N}^\star$ is a horizon and $\gamma \in (0, 1]$ a discount parameter. The transition kernels $p = (p_1, \ldots, p_H)$ and reward functions $r = (r_1, \ldots, r_H)$ can have distinct definitions in each step of the episode. The optimal value of selecting action $a$ in state $s_1$ is

$$Q^\star(s_1, a) = \max_\pi \mathbb{E}^\pi \left[ \sum_{h=1}^H \gamma^{h-1} r_h(s_h, a_h) \middle| a_1 = a \right],$$

where the supremum is taken over (deterministic) policies $\pi = (\pi_1, \ldots, \pi_H)$, and the expectation is on a trajectory $s_1, a_1, \ldots, s_h, a_h$ where $s_h \sim p_{h-1}(\cdot | s_{h-1}, a_{h-1})$ and $a_h = \pi_h(s_h)$ for $h \in [2, H]$. With this definition, an optimal action in state $s_1$ is $a^\star \in \operatorname{argmax}_{a \in \mathcal{A}(s_1)} Q^\star(s_1, a)$.

We assume that there is a maximal number $K$ of actions available in each state, and that, for each $(s, a)$, the support of $p_h(\cdot | s, a)$ is bounded by $B$: that is, $B$ is the maximum number of possible next states when applying any action. We further assume that the rewards are bounded in $[0, 1]$. For each pair of integers $i, h$ such that $i \le h$, we introduce the notation $[i, h] = \{i, \ldots, h\}$ and $[h] = [1, h]$.

**$(\varepsilon, \delta)$-correct planning**    A sequential planning algorithm proceeds as follows. In each episode $t$, the agent uses a deterministic policy on the form $\pi^t = (\pi_1^t, \ldots, \pi_H^t)$ to generate a trajectory $(s_1, a_1^t, r_1^t, \ldots, s_H^t, a_H^t, r_H^t)$, where $a_h^t = \pi_h^t(s_h^t)$, $r_h^t$ is a reward with expectation $r_h(s_h^t, a_h^t)$ and $s_{h+1}^t \sim p_h(\cdot | s_h^t, a_h^t)$. After each episode the agent decides whether it should perform a new episode to refine its guess for a near-optimal action, or whether it can stop and make a guess. We denote by $\tau$ the stopping rule of the agent, that is the number of episodes performed, and $\hat{a}_\tau$ the guess.

We aim to build an $(\varepsilon, \delta)$-correct algorithm, that is an algorithm that outputs a guess $\hat{a}_\tau$ satisfying

$$\mathbb{P}\left(Q^\star(s_1, \hat{a}_\tau) > Q^\star(s_1, a^\star) - \varepsilon\right) \ge 1 - \delta \quad \Leftrightarrow \quad \mathbb{P}\left(\bar{r}_{(H\tau)}(\hat{a}_\tau) \le \varepsilon\right) \ge 1 - \delta \qquad (1)$$

while using as few calls to the generative model $n = H\tau$ (i.e. as few episodes $\tau$) as possible.

Our setup permits to propose algorithms for planning in the undiscounted episodic case (in which our bounds will not blow up when $\gamma = 1$) and in discounted MDPs with infinite horizon. Indeed, choosing $H$ such that $2\gamma^H/(1 - \gamma) \le \varepsilon$, an $(\varepsilon, \delta)$-correct algorithm for the discounted episodic setting recommends an action that is $2\varepsilon$-optimal for the discounted infinite horizon setting.

**A (recursive) baseline** Sparse Sampling [19] can be tuned to output a guess $\hat{a}$ that satisfies (1), as specified in the following lemma, which provides a baseline for our undiscounted episodic setting (see Appendix F). Note that Sparse Sampling is not strictly sequential as it does not repeatedly select trajectories. However, it can still be implemented using a forward model by storing states on a stack.

**Lemma 1.** *If $B < \infty$, Sparse Sampling using horizon $H$ and performing $\mathcal{O}\left((H^5/\varepsilon^2)\log(BK/\delta)\right)$ transitions in each node is $(\varepsilon, \delta)$-correct with sample complexity $O(n_{SS})$ for $n_{SS} := H^5(BK)^H/\varepsilon^2$.*

**Structure of the optimal Q-value function** In our algorithm, we will build estimates of the intermediate Q-values, that are useful to compute the optimal Q-value function $Q^\star(s_1, a)$. Defining

$$Q_h(s_h, a_h) = \max_\pi \mathbb{E}^\pi \left[ \sum_{i=h}^H \gamma^{i-h} r(s_i, a_i) \Bigg| s_h, a_h \right],$$

$Q^\star(s_1, a) = Q_1(s_1, a)$ and the optimal action-values $Q = (Q_1, \ldots, Q_H)$ can be computed recursively using the Bellman equations, where we use the convention $Q_{H+1}(\cdot, \cdot) = 0$:

$$Q_h(s_h, a_h) = r_h(s_h, a_h) + \gamma \sum_{s'} p_h(s'|s_h, a_h) \max_{a'} Q_{h+1}(s', a'), \quad h \in [H].$$

Let $\pi^\star = (\pi_1^\star, \ldots, \pi_H^\star)$ denote a deterministic optimal policy where, for $h \in [H]$, $\pi_h^\star(s_h) = \arg\max_a Q_h(s_h, a)$, with ties arbitrarily broken. Hence the optimal value in $s_h$ is $Q_h(s_h, \pi_h^\star(s_h))$.

# 3    The `MDP-GapE` Algorithm

In this section we present `MDP-GapE`, a generalization of `UGapE` [10] to Monte-Carlo planning. Like BAI-MCTS for games [18] a core component is the construction of confidence intervals on $Q_1(s_1, a)$. The construction below generalizes that of OP-MDP [3] for known transition probabilities.

**Confidence bounds on the $Q$-values** Our algorithm maintains empirical estimates, superscripted with the episode $t$, of the transition kernels $p$ and expected rewards $r$, which are assumed unknown.

Let $n_h^t(s_h, a_h, s_{h+1}) := \sum_{s=1}^t \mathbb{1}\left((s_h^s, a_h^s, s_{h+1}^s) = (s_h, a_h, s_{h+1})\right)$ be the number of observations of transition $(s_h, a_h, s_{h+1})$, and $R_h^t(s_h, a_h) := \sum_{s=1}^t r_h^s(s_h, a_h)\mathbb{1}\left((s_h^s, a_h^s) = (s_h, a_h)\right)$ the sum of rewards obtained when selecting $a_h$ in $s_h$. We define the empirical transition probabilities $\hat{p}^t$ and expected rewards $\hat{r}^t$ as follows, for state-action pairs such that $n_h^t(s_h, a_h) := \sum_s n_h^t(s_h, a_h, s) > 0$:

$$\hat{p}_h^t(s_{h+1}|s_h, a_h) := \frac{n_h^t(s_h, a_h, s_{h+1})}{n_h^t(s_h, a_h)}, \quad \text{and} \quad \hat{r}_h^t(s_h, a_h) := \frac{R_h^t(s_h, a_h)}{n_h^t(s_h, a_h)}.$$

As rewards are bounded in $[0, 1]$, we define the following Kullback-Leibler upper and lower confidence bounds on the mean rewards $r_h(s_h, a_h)$ [4]:

$$u_h^t(s_h, a_h) := \max\left\{ v : \mathrm{kl}\left(\hat{r}_h^t(s_h, a_h), v\right) \leq \frac{\beta^r(n_h^t(s_h, a_h), \delta)}{n_h^t(s_h, a_h)} \right\},$$

$$\ell_h^t(s_h, a_h) := \min\left\{ v : \mathrm{kl}\left(\hat{r}_h^t(s_h, a_h), v\right) \leq \frac{\beta^r(n_h^t(s_h, a_h), \delta)}{n_h^t(s_h, a_h)} \right\},$$

where $\beta^r$ is an exploration function and $\mathrm{kl}(u, v)$ is the binary Kullback-Leibler divergence between two Bernoulli distributions $\mathcal{B}\mathrm{er}(u)$ and $\mathcal{B}\mathrm{er}(v)$: $\mathrm{kl}(u, v) = u\log\frac{u}{v} + (1-u)\log\frac{1-u}{1-v}$. We adopt the convention that $u_h^t(s_h, a_h) = 1, \ell_h^t(s_h, a_h) = 0$ when $n_h^t(s_h, a_h) = 0$.

In order to define confidence bounds on the values $Q_h$, we introduce a confidence set on the probability vector $p_h(\cdot|s_h, a_h)$. We define $\mathcal{C}_h^t(s_h, a_h) = \Sigma_B$ if $n_h^t(s_h, a_h) = 0$ and otherwise

$$\mathcal{C}_h^t(s_h, a_h) := \left\{ p \in \Sigma_B : \mathrm{KL}\left(\hat{p}_h^t(\cdot|s_h, a_h), p\right) \leq \frac{\beta^p(n_h^t(s_h, a_h), \delta)}{n_h^t(s_h, a_h)} \right\},$$

where $\Sigma_B$ is the set of probability distribution over $B$ elements, $\beta^p$ is an exploration function and $\mathrm{KL}(p, q) = \sum_{s \in \mathrm{Supp}(p)} p(s)\log\frac{p(s)}{q(s)}$ is the Kullback-Leibler divergence between two categorical distributions $p$ and $q$ with supports satisfying $\mathrm{Supp}(p) \subseteq \mathrm{Supp}(q)$.

We now define our confidence bounds on the action values inductively. We use the convention $U_{H+1}^t(\cdot, \cdot) = L_{H+1}^t(\cdot, \cdot) = 0$, and for all $h \in [H]$,

$$U_h^t(s_h, a_h) = u_h^t(s_h, a_h) + \gamma \max_{p \in \mathcal{C}_h^t(s_h, a_h)} \sum_{s'} p(s'|s_h, a_h) \max_{a'} U_{h+1}^t(s', a'),$$

$$L_h^t(s_h, a_h) = \ell_h^t(s_h, a_h) + \gamma \min_{p \in \mathcal{C}_h^t(s_h, a_h)} \sum_{s'} p(s'|s_h, a_h) \max_{a'} L_{h+1}^t(s', a').$$

As explained in Appendix A of [9], optimizing over these KL confidence sets can be reduced to a linear program with convex constraints, that can be solved efficiently using Newton's method, which has complexity $O(B \log(d))$ where $d$ is the desired digit precision.

We provide in Section 4.1 an explicit choice for the exploration functions $\beta^r(n, \delta)$ and $\beta^p(n, \delta)$ that govern the size of the confidence intervals. Note that if the rewards or transitions are deterministic, or if we know $p$, we can adapt our confidence bounds by setting $\beta^p = 0$ or $\beta^r = 0$.

**MDP-GapE** As any fixed-confidence algorithm, `MDP-GapE` depends on the tolerance parameter $\varepsilon$ and the risk parameter $\delta$. The dependency in $\varepsilon$ is explicit in the stopping rule (4), while the dependency in $\delta$ is in the tuning of the confidence bounds, that depend on $\delta$.

After $t$ trajectories observed, `MDP-GapE` selects the $(t + 1)$-st trajectory using the policy $\pi^{t+1} = (\pi_1^{t+1}, \dots, \pi_H^{t+1})$ where the first action choice is made according to `UGapE`:

$$\pi_1^{t+1}(s_1) = \underset{b \in \{b^t, c^t\}}{\operatorname{argmax}} \left[ U_1^t(s_1, b) - L_1^t(s_1, b) \right],$$

where $b^t$ is the current guess for the best action, which is the action $b$ with the smallest upper confidence bound on its gap $Q_1^\star(s_1, a^\star) - Q_1(s_1, b)$, and $c^t$ is some challenger:

$$b^t = \underset{b}{\operatorname{argmin}} \left[ \max_{a \neq b} U_1^t(s_1, a) - L_1^t(s_1, b) \right], \tag{2}$$

$$c^t = \underset{c \neq b^t}{\operatorname{argmax}} U_1^t(s_1, c). \tag{3}$$

Then for all remaining steps we follow an optimistic policy, for all $h \in [2, H]$,

$$\pi_h^{t+1}(s_h) = \underset{a}{\operatorname{argmax}} U_h^t(s_h, a).$$

The stopping rule of `MDP-GapE` is

$$\tau = \inf\{t \in \mathbb{N} : U_1^t(s_1, c^t) - L_1^t(s_1, b^t) \leq \varepsilon\}, \tag{4}$$

and the guess output when stopping is $\hat{a}_\tau = b^\tau$. A generic implementation of `MDP-GapE` is given in Algorithm 1 in Appendix A, where we also discuss some implementation details. Note that, in sharp contrast with the deterministic stopping rule proposed for Sparse Sampling in Lemma 1, `MDP-GapE` uses an adaptive stopping rule.

The high-level intuition behind `MDP-GapE` is that unlike a greedy optimistic policy, `MDP-GapE` does not attempt to minimize regret while learning the best action to take in step 1. The UGapE policy followed at depth 1 indeed explores much more than a purely optimistic algorithm. At depths larger than 1, however, `MDP-GapE` does follow a greedy optimistic policy. This combination of policy choices is crucial for the theoretical analysis of the algorithm, and for quickly achieving the proposed stopping condition (4): stop when one of the confidence intervals on the value at depth 1 is larger than and separated from the others.

# 4  Analysis of `MDP-GapE`

Recall that `MDP-GapE` uses policy $\pi^{t+1} = (\pi_1^{t+1}, \dots, \pi_H^{t+1})$ to select the $(t + 1)$-st trajectory, $s_1, a_1^{t+1}, s_2^{t+1}, a_2^{t+1}, \dots, s_H^{t+1}, a_H^{t+1}$, satisfying $a_h^{t+1} = \pi_h^{t+1}(s_h^{t+1})$ and $s_{h+1}^{t+1} \sim p_h \left( \cdot \left| s_h^{t+1}, a_h^{t+1} \right. \right)$.

**High probability event** To define an event $\mathcal{E}$ that holds with high probability, let $\mathcal{E}^r$ (resp. $\mathcal{E}^p$) be the event that the confidence regions for the mean rewards (resp. transition kernels) are correct:

$$\mathcal{E}^r := \left\{\forall t \in \mathbb{N}^*, \forall h \in [H], \forall (s_h, a_h) \in \mathcal{S} \times \mathcal{A} : \ r_h(s_h, a_h) \in \left[\ell_h^t(s_h, a_h), u_h^t(s_h, a_h)\right]\right\},$$
$$\mathcal{E}^p := \left\{\forall t \in \mathbb{N}^*, \forall h \in [H], \forall (s_h, a_h) \in \mathcal{S} \times \mathcal{A} : \ p_h(\cdot|s_h, a_h) \in \mathcal{C}_h^t(s_h, a_h)\right\}.$$

For a state-action pair $(s_h, a_h)$, let $p_h^\pi(s_h, a_h)$ be the probability of reaching it at step $h$ under policy $\pi$, and let $p_h^t(s_h, a_h) = p_h^{\pi^t}(s_h, a_h)$. We define the *pseudo-counts* of the number of visits of $(s_h, a_h)$ as $\bar{n}_h^t(s_h, a_h) := \sum_{s=1}^t p_h^s(s_h, a_h)$. As $n_h^t(s_h, a_h) - \bar{n}_h^t(s_h, a_h)$ is a martingale, the counts should not be too far from the pseudo-counts. Given a rate function $\beta^{\mathrm{cnt}}$, we define the event

$$\mathcal{E}^{\mathrm{cnt}} := \left\{\forall t \in \mathbb{N}^\star, \forall h \in [H], \forall (s_h, a_h) \in \mathcal{S} \times \mathcal{A} : \ n_h^t(s_h, a_h) \geq \frac{1}{2}\bar{n}_h^t(s_h, a_h) - \beta^{\mathrm{cnt}}(\delta)\right\}.$$

Finally, we define $\mathcal{E}$ to be the intersection of these three events: $\mathcal{E} = \mathcal{E}^r \cap \mathcal{E}^p \cap \mathcal{E}^{\mathrm{cnt}}$.

## 4.1 Correctness

One can easily prove by induction (see Appendix B) that

$$\mathcal{E}^r \cap \mathcal{E}^p \subseteq \bigcap_{t \in \mathbb{N}^\star} \bigcap_{h=1}^H \left[\bigcap_{s_h, a_h} \left(Q_h(s_h, a_h) \in \left[L_h^t(s_h, a_h), U_h^t(s_h, a_h)\right]\right)\right].$$

As the arm $\hat{a}$ output by `MDP-GapE` satisfies $L_1(s_1, \hat{a}) > \max_{c \neq \hat{a}} U_1(s_1, c) - \varepsilon$, on the event $\mathcal{E} \subseteq \mathcal{E}^r \cap \mathcal{E}^p$ it holds that $Q_1(s_1, \hat{a}) > \max_{c \neq \hat{a}} Q_1(s_1, c) - \varepsilon$. Thus `MDP-GapE` can only output an $\varepsilon$-optimal action. Hence a sufficient condition for `MDP-GapE` to be $(\varepsilon, \delta)$-correct is $\mathbb{P}(\mathcal{E}) \geq 1 - \delta$.

In Lemma 2 below, we provide a calibration of the thresholds functions $\beta^r, \beta^p$ and $\beta^{\mathrm{cnt}}$ such that this sufficient condition holds. This result, proved in Appendix C, relies on new time-uniform concentration inequalities that follow from the method of mixtures [7].

**Lemma 2.** *For all $\delta \in [0, 1]$, it holds that $\mathbb{P}(\mathcal{E}) \geq 1 - \delta$ for the choices*

$$\beta^r(n, \delta) = \log(3(BK)^H/\delta) + \log\left(e(1+n)\right), \quad \beta^{\mathrm{cnt}}(\delta) = \log\left(3(BK)^H/\delta\right),$$
$$\text{and} \quad \beta^p(n, \delta) = \log\left(3(BK)^H/\delta\right) + (B-1)\log\left(e(1+n/(B-1))\right).$$

*Moreover, the maximum of these three thresholds defined (by continuity when $B = 1$) as*

$$\beta(n, \delta) := \max_{c \in \{r, p, \mathrm{cnt}\}} \beta^c(n, \delta) = \log\left(3(BK)^H/\delta\right) + (B-1)\log\left(e(1+n/(B-1))\right),$$

*is such that $n \mapsto \beta(n, \delta)$ is non-decreasing and $n \mapsto \beta(n, \delta)/n$ is non-increasing.*

## 4.2 Sample Complexity

In order to state our results, we define the following sub-optimality gaps. $\Delta_h(s_h, a_h)$ measures the gap in future discounted reward between the optimal action $\pi_h^\star(s_h)$ and the action $a_h$, whereas $\Delta_1^\star(s_1, a_1)$ also takes into account the gap of the second best action and the tolerance level $\varepsilon$.

**Definition 1.** *Recall that $\Delta = \min_{a \neq a^\star}\left[Q_1(s_1, a^\star) - Q_1(s_1, a)\right]$. For all $h \in [H]$, we let*

$$\Delta_h(s_h, a_h) = Q_h(s_h, \pi_h^\star(s_h)) - Q_h(s_h, a_h),$$
$$\Delta_1^\star(s_1, a_1) = \max\left(\Delta_1(s_1, a_1); \Delta; \varepsilon\right),$$

*and we denote* $\tilde{\Delta}_h(s_h, a_h) = \begin{cases} \Delta_1^\star(s_h, a_h), & \text{if } h = 1, \\ \Delta_h(s_h, a_h), & \text{if } h \geq 2. \end{cases}$

Our sample complexity bounds follow from the following crucial theorem, which we prove in Appendix D, that relates the pseudo-counts of state-action pairs at time $\tau$ to the corresponding gap.

**Theorem 1.** *If $\mathcal{E}$ holds, every $(s_h, a_h)$ is such that*

$$\bar{n}_h^\tau(s_h, a_h)\tilde{\Delta}_h(s_h, a_h) \leq 64\sqrt{2}(1+\sqrt{2})\left(\sqrt{BK}\right)^{H-h}\sqrt{\bar{n}_h^\tau(s_h, a_h)\beta(\bar{n}_h^\tau(s_h, a_h), \delta)}.$$

Introducing the constant $C_0 = (64\sqrt{2}(1 + \sqrt{2}))^2$ and letting $c_\delta = \log\left(\frac{3(BK)^H}{\delta}\right)$, Lemma 12 stated in Appendix G permits to prove that, on the event $\mathcal{E}$, any $(s_h, a_h)$ for which $\tilde{\Delta}_h(s_h, a_h) > 0$ satisfies

$$\bar{n}_h^\tau(s_h, a_h) \leq \frac{C_0(BK)^{H-h}}{\tilde{\Delta}_h^2(s_h, a_h)}\left[c_\delta + 2(B-1)\log\left(\frac{C_0(BK)^{H-h}}{\tilde{\Delta}_h^2(s_h, a_h)}\left[\frac{c_\delta}{\sqrt{B-1}} + 2\sqrt{e(B-1)}\right]\right) + (B-1)\right] \quad (5)$$

As $\tilde{\Delta}_1(s_1, a_1) = \max(\Delta_1(s_1, a_1); \Delta; \varepsilon)$ is positive, the following corollary follows from summing the inequality over $a_1$, as $\bar{n}_1^\tau(s_1, a_1) = n_1^\tau(s_1, a_1)$ and $\tau = \sum_{a_1} n_1^\tau(s_1, a_1)$.

**Corollary 1.** *The number of episodes used by MDP-GapE satisfies*

$$\mathbb{P}\left(\tau = \mathcal{O}\left(\sum_{a_1} \frac{(BK)^{H-1}}{(\Delta_1(s_1, a_1) \vee \Delta \vee \varepsilon)^2}\left[\log\left(\frac{1}{\delta}\right) + BH\log(BK)\right]\right)\right) \geq 1 - \delta \ .$$

The upper bound on the sample complexity $n = H\tau$ of `MDP-GapE` that follows from Corollary 1 improves over the $\mathcal{O}(H^5(BK)^H/\varepsilon^2)$ sample complexity of `Sparse Sampling`. It is also smaller than the $\mathcal{O}(H^4(BK)^H/\Delta^2)$ samples needed for `BRUE` to have a reasonable upper bound on its simple regret. The improvement is twofold: first, this new bound features the problem dependent gap $\Delta(s_1, a_1) \vee \Delta \vee \varepsilon$ for each action $a_1$ in state $s_1$, whereas previous bounds were only expressed with $\varepsilon$ or $\Delta$. Second, it features an improved scaling in $H^2$.

It is also possible to provide bounds that features the gaps $\tilde{\Delta}_h(s_h, a_h)$ in the *whole* tree, beyond depth one. To do so, we shall consider *trajectories* $t_{1:H} = (s_1, a_1, \ldots, s_H, a_H)$ or trajectory *prefixes* $t_{1:h} = (s_1, a_1, \ldots, s_h, a_h)$ for $h \in [H]$. Introducing the probability $p_h^\pi(t_{1:h})$ that the prefix $t_{1:h}$ is visited under policy $\pi$, we can further define the pseudo-counts $\bar{n}_h^t(t_{1:h}) = \sum_{s=1}^t p_h^{\pi^s}(t_{1:h})$. One can easily show that for all $h \in [H]$, $\bar{n}_H^\tau(t_{1:H}) \leq \bar{n}_h^\tau(t_{1:h}) \leq \bar{n}_h^\tau(s_h, a_h)$, if $(s_h, a_h)$ is the state-action pair visited in step $h$ in the trajectory $t_{1:H}$, and (5) leads to the following upper bound.

**Corollary 2.** *On the event $\mathcal{E}$, $\bar{n}_h^\tau(t_{1:h}) = \mathcal{O}\left(\left[\min_{\ell=1}^h \frac{(BK)^{H-\ell}}{(\tilde{\Delta}_\ell(s_\ell, a_\ell))^2}\right]\log\left(\frac{3(BK)^H}{\delta}\right)\right)$.*

In particular, using that $\tau = \sum_{t_{1:H} \in \mathcal{T}} \bar{n}_h^\tau(t_{1:H})$ where $\mathcal{T}$ is the set of $(BK)^H$ complete trajectories leads to a sample complexity bound featuring all gaps. However, its improvement over the bound of Corollary 1 is not obvious in the general case. For $B = 1$, that is for planning in a deterministic MDP with possibly random rewards, a slightly different proof technique leads to the following improved gap-dependent sample complexity bound (see the proof in Appendix E).

**Theorem 2** (deterministic case). *When $B = 1$, `MDP-GapE` satisfies*

$$\mathbb{P}\left(\tau = \mathcal{O}\left(\sum_{t_{1:H} \in \mathcal{T}} \left[\min_{h=1}^H \frac{H\left(\sum_{\ell=h}^H \gamma^\ell\right)^2}{\left(\tilde{\Delta}_h^2(s_h, a_h)\right)^2}\right]\left(\log\left(\frac{1}{\delta}\right) + H\log(K)\right)\right)\right) \geq 1 - \delta.$$

**Scaling in $\varepsilon$**  A majority of prior work on planning in MDPs has obtained sample complexity bounds that scale with $\varepsilon$ only, in the discounted setting. Neglecting the gaps, Corollary 1 gives a $\mathcal{O}(H^2(BK)^H/\varepsilon^2)$ upper bound that yields a crude $\tilde{\mathcal{O}}\left(\varepsilon^{-[2+\log(BK)/\log(1/\gamma)]}\right)$ sample complexity in the discounted setting in which $H \sim \log(1/\varepsilon)/\log(1/\gamma)$. This exponent is larger than that in previous work, which features some notion of near-optimality dimension $\kappa$ (see Table 1). However, our analysis was not tailored to optimizing this exponent, and we show in Section 5 that the empirical scaling of `MDP-GapE` in $\varepsilon$ can be much smaller than the one prescribed by the above crude bound.

**Lower bounds**  To the best of our knowledge, the only available lower bound on the sample complexity of MCTS planning in general MDPs is the $(1/\varepsilon)^{1/\log(1/\gamma)}$ worst-case bound given by Kearns et al. [19], which is proved using an MDP that is a binary tree ($B < \infty$). In the open-loop setting, Bubeck and Munos [2] prove a minimax lower bound that is $\Omega\left(\varepsilon^{-\log K/\log(1/\gamma)}\right)$ if $\gamma\sqrt{K} > 1$ and $\Omega\left(\varepsilon^{-2}\right)$ if $\gamma\sqrt{K} \leq 1$. As for problem-dependent results, the only available results hold for $H = 1$, for which MCTS planning boils down to finding an arm with mean that is within $\varepsilon$ of the best mean in a bandit model. In that case, the lower bound of Mannor and Tsitsiklis [22] indeed features the gaps at depth-one that appear in Corollary 1. Deriving problem-dependent lower bound for $H \geq 2$ is left as an important future work.

# 5 Numerical Experiments[1]

We consider random discounted MDPs with infinite horizon in which the maximal number $B$ of successor states and the sparsity of rewards are controlled. The transition kernel is generated as follows: for each transition in $\mathcal{S} \times \mathcal{A}$, we uniformly pick $B$ next states in $\mathcal{S}$. The cumulative transition probabilities to these states are computed by sorting $B-1$ numbers uniformly sampled in $(0, 1)$. The reward kernel is computed by selecting a proportion of the transitions to have non-zero rewards with means sampled uniformly in $(0, 1)$. The values for these parameters are shown in Table 3a.

Table 3: Experimental setting.

| (a) Environment parameters | | (b) `MDP-GapE` parameters | |
|---|---|---|---|
| States $\mathcal{S}$ | $10^5$ | Discount factor $\gamma$ | 0.7 |
| Actions $\mathcal{A}$ | 5 | Confidence level $\delta$ | 0.1 |
| Number $B$ of successors | 2 | Exploration function $\beta_r(n_h^t, \delta)$ | $\log \frac{1}{\delta} + \log n_h^t$ |
| Reward sparsity | 0.5 | Exploration function $\beta_p(n_h^t, \delta)$ | $\log \frac{1}{\delta} + \log n_h^t$ |

The main objective of our numerical experiments is to empirically verify several properties of `MDP-GapE`, but we acknowledge that these experiments have some limitations. Planning algorithms such as `MDP-GapE` are usually intended for the case $(BK)^{H-1} \ll SA$, which does not hold in our experiments (despite the large state space). Moreover, we use tighter threshold functions than those prescribed by theory, as is sometimes done in the bandit literature. These choices of thresholds are still inspired by our theoretical results, for their scaling in $n_h^t(s, a)$, un-doing a few union bounds that were found to be conservative in practice.

**Fixed-confidence: Correction and sample complexity**   We verify empirically that `MDP-GapE` is $(\varepsilon, \delta)$-correct while stopping with a reasonable number of oracle calls. Table 3b shows the choice of parameters for the algorithm. For various values of the desired accuracy $\varepsilon$ and of the corresponding planning horizon $H = \lceil \log_\gamma(\varepsilon(1-\gamma)/2) \rceil$ (see Section 2), we run simulations on 200 random MDPs. We report in Table 4 the distribution of the number $n = \tau H$ of oracle calls and the simple regret $\bar{r}_n(\hat{a}_n)$ of `MDP-GapE` over these 200 runs. We first observe that `MDP-GapE` satisfies $\bar{r}_n(\hat{a}_n) < \varepsilon$ in all simulations, despite the use of smaller exploration functions compared to those prescribed in Lemma 2. We then compare the empirical sample complexity of `MDP-GapE` to the number of samples that Sparse Sampling would use. The sample complexity of Sparse Sampling with parameter $C$ (number of calls to the generative model in each node) is $(K^{H+1} - K)/(K - 1)$ for $C = 1$ and of order $\sum_{h=0}^{H-1}[(KC) \times (K(\min(B, C)))^h]$ for larger values of $C$. Thus, beyond very small $C$, the runtime of SS is prohibitively too large to try the algorithm in our setting (larger than $(BK)^H = 10^H$). For $C = 1$, the sample complexity of SS is $2.0 \times 10^4$, $4.9 \times 10^5$ and $1.2 \times 10^7$ in the 3 experiments in Table 4, which is larger than the maximal sample complexity observed for MDP-GapE.

Table 4: Simple regret and number of oracle calls, collected on 200 simulations

| $\varepsilon$ | $H$ | MDP-GapE | | |
|---|---|---|---|---|
| | | max $r_n$ | median $n$ | max $n$ |
| 1 | 6 | $3.6 \times 10^{-2}$ | $8.6 \times 10^3$ | $1.8 \times 10^4$ |
| 0.5 | 8 | $5.2 \times 10^{-3}$ | $7.3 \times 10^4$ | $2.0 \times 10^5$ |
| 0.2 | 10 | 0 | $5.0 \times 10^5$ | $2.3 \times 10^6$ |

**Scaling in $\varepsilon$**   As discussed above, Corollary 1 with the aforementioned choice of the planning horizon, yields a crude sample complexity bound of order $\tilde{\mathcal{O}}\left(\varepsilon^{-[2+\log(BK)/\log(1/\gamma)]}\right) = \tilde{\mathcal{O}}\left((1/\varepsilon)^{8.4}\right)$ in our experimental setting. However, we observe that the empirical exponent can be much smaller in practice. To see that, we plot in in log-log scale in Figure 1 the sample complexity $n$ as a function of $1/(\Delta \vee \varepsilon)$ when running `MDP-GapE` for 5 different values of $\varepsilon$ and 200 random MDPs for each

value (each dot correponds to one value of $\varepsilon$ and one MDP). The 5 vertical groups of dots correspond to the 5 values of $\varepsilon$ and to MDPs for which $\Delta \vee \varepsilon = \varepsilon$. In particular, by measuring the slope of the curve we obtain that the maximal sample complexity among those MDPs scales in $n \simeq \mathcal{O}\big((1/\varepsilon)^{3.0}\big)$. Dots that are between the vertical groups correspond to MDPs for which the smallest gap $\Delta$ was larger than $\varepsilon$, and for which the sample complexity is typically smaller than this worst-case value.

**Comparison to the state of the art**   In the fixed-confidence setting, most existing algorithms are considered theoretical and cannot be applied to practical cases. For instance, for our problem with $K = 5$ and $\varepsilon = 1$, Sparse Sampling [19] and SmoothCruiser [14] both require a fixed budget[2] of at least $n_{\text{SS}} = 8 \times 10^9$. Likewise, Trailblazer [13] is a recursive algorithm which did not terminate in our setting. We did not implement StOP [28] as it requires to store a tree of policies, which is very costly even for moderate horizons. In comparison, Table 4 shows that MDP-GapE stopped after $n = 1.8 \times 10^4$ oracle calls in the worst case. To the best of our knowledge, MDP-GapE is the first $(\varepsilon, \delta)$-correct algorithm for general MDPs with an easy implementation and a reasonable running time in practice. The only planning algorithms that can be run in practice are in the fixed-budget setting, which we now consider.

**Fixed-budget evaluation**   We compare MDP-GapE to three existing baselines: first, the KL-OLOP algorithm [21], which uses the same upper-confidence bounds on the rewards $u_h^t$ and states values $U_h^t$ as MDP-GapE, but is restricted to *open-loop* policies, i.e. sequences of actions only. Second, the BRUE algorithm [8] which explores uniformly and handles closed-loop policies. Third, the popular UCT algorithm [20], which is also closed-loop and performs optimistic exploration at all depths. UCT and its variants lack theoretical guarantees, but they have been shown successful empirically in many applications. For each algorithm, we tune the planning horizon $H$ similarly to KL-OLOP, by dividing the available budget $n$ into $\tau$ episodes, where $\tau$ is the largest integer such that $\tau \log \tau / (2 \log 1/\gamma) \le n$, and choose $H = \log \tau / (2 \log 1/\gamma)$. The exploration functions are those of KL-OLOP and depend on $\tau$: $\beta_r(n_h^t, \delta) = \beta_p(n_h^t, \delta) = \log(\tau)$. Again, we perform 200 simulations and report in Figure 2 the mean simple regret, along with its $95\%$ confidence interval. We observe that MDP-GapE compares favourably with these baselines in the high-budget regime.

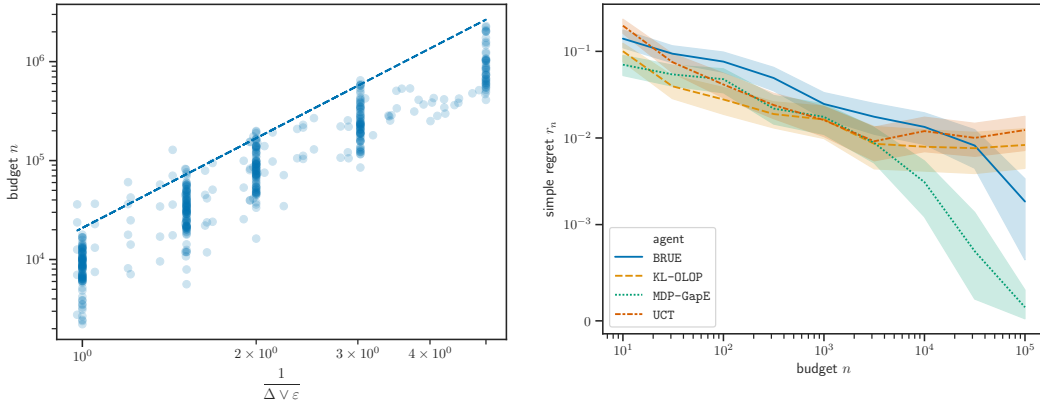

Figure 1: Dependency of the maximum number $n$ of oracle calls with respect to $1/(\Delta \vee \varepsilon)$.

Figure 2: Comparison to other planning algorithms in a fixed-budget setting.

## 6   Conclusion

We proposed a new, efficient algorithm for Monte-Carlo planning in Markov Decision Processes, that combines tools from best arm identification and optimistic planning and exploits tight confidence regions on mean rewards and transitions probabilities. We proved that MDP-GapE attains the smallest existing gap-dependent sample complexity bound for general MDPs with stochastic rewards and transitions, when the branching factor $B$ is finite. In future work, we will investigate the worst-case complexity of MDP-GapE, that is try to derive an upper bound on its sample complexity that only features $\varepsilon$ and some appropriate notion of near-optimality dimension.

## Broader Impact

Monte-Carlo Tree Search methods are very popular but their theoretical understanding remains limited. This work propose new sample complexity bounds for an efficient MCTS algorithm, which can be interesting for both theoreticians and practioners. However, this paper was not targeted towards a particular application, so a wider broader impact discussion is not applicable.

## Acknowledgments and Disclosure of Funding

We acknowledge the support of the European CHIST-ERA project DELTA and the French ANR project BOLD (ANR-19-CE23-0026-04). Anders Jonsson is partially supported by the Spanish grants TIN2015-67959 and PCIN-2017-082.

## Footnotes

[1]The source code of our experiments is available at https://eleurent.github.io/planning-gap-complexity/

[2]In non-regularized MDPs, SmoothCruiser has the same sample complexity as Sparse Sampling.

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
