[Supplementary Material]

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

# A   Detailed Algorithm

In this section we provide a detailed algorithm for `MDP-GapE`, namely Algorithm 1.

---

**Algorithm 1** `MDP-GapE`

---

1: **Input:** confidence level $\delta$, tolerance $\varepsilon$
2: initialize data lists $\mathcal{D}_h \leftarrow [\,]$ for all $h \in [H]$
3: **for** $t = 1 \ldots$ **do**
4:    //Update confidence bounds
5:    $U_h^{t-1}, L_h^{t-1} \leftarrow \texttt{UpdateBounds}(t, \delta, \mathcal{D}_h)$
6:    **if** $U_1^{t-1}(s_1, c^t) - L_1^{t-1}(s_1, b^t) \leq \varepsilon$ **then**
7:       **return** $b_{t-1}$, **break**
8:    **end if**
9:    // Best
10:    $b^{t-1} \leftarrow \underset{b}{\operatorname{argmin}} \left[ \max_{a \neq b} U_1^{t-1}(s_1, a) - L_1^{t-1}(s_1, b) \right]$
11:    //Challenger
12:    $c^{t-1} \leftarrow \underset{c \neq b^t}{\operatorname{argmax}} \, U_1^{t-1}(s_1, c)$
13:    //Exploration
14:    $a_1^t \leftarrow \underset{a \in \{b^{t-1}, c^{t-1}\}}{\operatorname{argmax}} \left[ U_1^{t-1}(s_1, a) - L_1^{t-1}(s_1, a) \right]$
15:    observe reward $r_1^t$, next state $s_2^t$, save $\mathcal{D}_1$.append$(s_1^t, a_1^t, s_2^t, r_1^t)$
16:    **for** step $h = 2, \ldots, H$ **do**
17:       $a_h^t \leftarrow \underset{a}{\operatorname{argmax}} \, U_h^{t-1}(s_h^t, a)$
18:       observe reward $r_{h-1}^t$, next state $s_h^t$, save $\mathcal{D}_h$.append$(s_h^t, a_h^t, s_{h+1}^t, r_h^t)$
19:    **end for**
20: **end for**

---

**Implementation details**    There are different ways to store and update the confidence bounds on the $Q$-value (that is, to specify the `UpdateBounds` subroutine) according to how we merge information across states.

The most obvious one, suggested by previous work [2, 21, 3] (and also implemented for our experiments) does not merge information at all and builds a search *tree* in which a node $(s_h, a_h)$ at depth $h$ is identified with the sequence of $h$ states and actions that leads to it. It leads to a very simple update: after each trajectory, one only needs to update the confidence bounds, $U_h(s_h, a_h)$ and $L_h(s_h, a_h)$, of the visited action-state pairs. Another option is to merge information for the same states and a fixed depth. But in this case the search tree becomes a *graph* and after each trajectory we need to re-compute the values $U_h(s_h, a_h)$ for all stored state action pairs $(s_h, a_h)$ at each depth.

# B   Correctness of `MDP-GapE`

In this section we prove the correctness of `MDP-GapE` under the assumption that the event $\mathcal{E}^r \cap \mathcal{E}^p$ holds. Concretely, we prove by induction that

$$\mathcal{E}^r \cap \mathcal{E}^p \subseteq \bigcap_{t \in \mathbb{N}^\star} \bigcap_{h=1}^{H} \left[ \bigcap_{s_h, a_h} \left( Q_h(s_h, a_h) \in \left[ L_h^t(s_h, a_h), U_h^t(s_h, a_h) \right] \right) \right].$$

The base case is given by $h = H + 1$, in which case by our previous convention,

$$L_{H+1}^t(\cdot, \cdot) = Q_{H+1}(\cdot, \cdot) = U_{H+1}^t(\cdot, \cdot) = 0.$$

For the inductive case, assume that the inclusion holds at depth $h + 1$. Then we have

$$
\begin{aligned}
L_h^t(s_h, a_h) &= \ell_h^t(s_h, a_h) + \gamma \min_{p \in \mathcal{C}_h^t(s_h, a_h)} \sum_{s'} p(s'|s_h, a_h) \max_{a'} L_{h+1}^t(s', a') \\
&\leq \ell_h^t(s_h, a_h) + \gamma \sum_{s'} p_h(s'|s_h, a_h) \max_{a'} L_{h+1}^t(s', a') \\
&\leq r_h(s_h, a_h) + \gamma \sum_{s'} p_h(s'|s_h, a_h) Q_{h+1}(s', \arg\max_{a'} L_{h+1}^t(s', a')) \\
&\leq r_h(s_h, a_h) + \gamma \sum_{s'} p_h(s'|s_h, a_h) Q_{h+1}(s', \pi_{h+1}^*(s')) = Q_h(s_h, a_h) \\
&\leq u_h^t(s_h, a_h) + \gamma \sum_{s'} p_h(s'|s_h, a_h) \max_{a'} U_{h+1}^t(s', a') \\
&\leq u_h^t(s_h, a_h) + \gamma \max_{p \in \mathcal{C}_h^t(s_h, a_h)} \sum_{s'} p(s'|s_h, a_h) \max_{a'} U_{h+1}^t(s', a') = U_h^t(s_h, a_h),
\end{aligned}
$$

where we have used $r_h(s_h, a_h) \in \left[ \ell_h^t(s_h, a_h), u_h^t(s_h, a_h) \right]$ and $p_h(\cdot|s_h, a_h) \in \mathcal{C}_h^t(s_h, a_h)$.

## C    Concentration Events

In this section we prove that the event $\mathcal{E}$ holds with high probability. But before we need several concentration inequalities.

### C.1    Deviation Inequality for Categorical Distributions

Let $X_1, X_2, \ldots, X_n, \ldots$ be i.i.d. samples from a distribution supported over $\{1, \ldots, m\}$, of probabilities given by $p \in \Sigma_m$, where $\Sigma_m$ is the probability simplex of dimension $m - 1$. We denote by $\widehat{p}_n$ the empirical vector of probabilities, i.e. for all $k \in \{1, \ldots, m\}$

$$
\widehat{p}_{n,k} = \frac{1}{n} \sum_{\ell=1}^{n} \mathbb{1}(X_\ell = k).
$$

Note that an element $p \in \Sigma_m$ will sometimes be seen as an element of $\mathbb{R}^{m-1}$ since $p_m = 1 - \sum_{k=1}^{m-1} p_k$. This should be clear from the context. We denote by $H(p)$ the (Shannon) entropy of $p \in \Sigma_m$,

$$
H(p) = \sum_{k=1}^{m} p_k \log(1/p_k).
$$

**Proposition 1.** *For all $p \in \Sigma_m$, for all $\delta \in [0, 1]$,*

$$
\mathbb{P}\left( \exists n \in \mathbb{N}^*, \, n \, \mathrm{KL}(\widehat{p}_n, p) > \log(1/\delta) + (m - 1) \log \left( e(1 + n/(m - 1)) \right) \right) \leq \delta.
$$

*Proof.* We apply the method of mixture with a Dirichlet prior on the mean parameter of the exponential family formed by the set of categorical distribution on $\{1, \ldots, m\}$. Letting

$$
\varphi_p(\lambda) = \log \mathbb{E}_{X \sim p} \left[ e^{\lambda X} \right] = \log(p_m + \sum_{k=1}^{m-1} p_k e^{\lambda_k}),
$$

be the log-partition function, the following quantity is a martingale:

$$
M_n^\lambda = e^{n \langle \lambda, \widehat{p}_n \rangle - n \varphi_p(\lambda)}.
$$

We set a Dirichlet prior $q \sim \mathcal{D}ir(\alpha)$ with $\alpha \in \mathbb{R}_+^{*\,m}$ and for $\lambda_q = (\nabla\varphi_p)^{-1}(q)$ and consider the integrated martingale

$$
\begin{aligned}
M_n &= \int M_n^{\lambda_q} \frac{\Gamma\left(\sum_{k=1}^m \alpha_k\right)}{\prod_{k=1}^m \Gamma(\alpha_k)} q_k^{\alpha_k - 1} \, \mathrm{d}q \\
&= \int e^{n\left(\mathrm{KL}(\widehat{p}_n, p) - \mathrm{KL}(\widehat{p}_n, q)\right)} \frac{\Gamma\left(\sum_{k=1}^m \alpha_k\right)}{\prod_{k=1}^m \Gamma(\alpha_k)} q_k^{\alpha_k - 1} \, \mathrm{d}q \\
&= e^{n\,\mathrm{KL}(\widehat{p}_n, p) + nH(\widehat{p}_n)} \int \frac{\Gamma\left(\sum_{k=1}^m \alpha_k\right)}{\prod_{k=1}^m \Gamma(\alpha_k)} q_k^{n\widehat{p}_{n,k} + \alpha_k - 1} \, \mathrm{d}q \\
&= e^{n\,\mathrm{KL}(\widehat{p}_n, p) + nH(\widehat{p}_n)} \frac{\Gamma\left(\sum_{k=1}^m \alpha_k\right)}{\prod_{k=1}^m \Gamma(\alpha_k)} \frac{\prod_{k=1}^m \Gamma(\alpha_k + n\widehat{p}_{n,k})}{\Gamma\left(\sum_{k=1}^m \alpha_k + n\right)},
\end{aligned}
$$

where in the second inequality we used Lemma 3. Now we choose the uniform prior $\alpha = (1, \ldots, 1)$. Hence we get

$$
\begin{aligned}
M_n &= e^{n\,\mathrm{KL}(\widehat{p}_n, p) + nH(\widehat{p}_n)} (m-1)! \frac{\prod_{k=1}^m \Gamma(1 + n\widehat{p}_{n,k})}{\Gamma(m+n)} \\
&= e^{n\,\mathrm{KL}(\widehat{p}_n, p) + nH(\widehat{p}_n)} (m-1)! \frac{\prod_{k=1}^m (n\widehat{p}_{n,k})!}{n!} \frac{n!}{(m+n-1)!} \\
&= e^{n\,\mathrm{KL}(\widehat{p}_n, p) + nH(\widehat{p}_n)} \frac{1}{\binom{n}{n\widehat{p}_n}} \frac{1}{\binom{m+n-1}{m-1}}.
\end{aligned}
$$

Thanks to Theorem 11.1.3 by [5] we can upper bound the multinomial coefficient as follows: for $M \in \mathbb{N}^*$ and $x \in \{0, \ldots, M\}^m$ such that $\sum_{k=1}^m x_k = M$ it holds

$$
\binom{M}{x} = \frac{M!}{\prod_{k=1}^m x_k!} \le e^{MH(x/M)}.
$$

Using this inequality we obtain

$$
\begin{aligned}
M_n &\ge e^{n\,\mathrm{kl}(\widehat{p}_n, p) + nH(\widehat{p}_n) - nH(\widehat{p}_n) - (m+n-1)H\left((m-1)/(m+n-1)\right)} \\
&= e^{n\,\mathrm{KL}(\widehat{p}_n, p) - (m+n-1)H\left((m-1)/(m+n-1)\right)}.
\end{aligned}
$$

It remains to upper-bound the entropic term

$$
\begin{aligned}
(m+n-1)H\left((m-1)/(m+n-1)\right) &= (m-1)\log\frac{m+n-1}{m-1} + n\log\frac{m+n-1}{n} \\
&\le (m-1)\log\left(1 + n/(m-1)\right) + n\log(1 + (m-1)/n) \\
&\le (m-1)\log\left(1 + n/(m-1)\right) + (m-1).
\end{aligned}
$$

Thus we can lower bound the martingale as follows

$$
M_n \ge e^{n\,\mathrm{KL}(\widehat{p}_n, p)} \left(e(1 + n/(m-1))\right)^{m-1}.
$$

Using the fact that, for any supermartingale it holds that

$$
\mathbb{P}\left(\exists n \in \mathbb{N}^* : M_n > 1/\delta\right) \le \delta \mathbb{E}[M_1], \tag{6}
$$

which is a well-known property used in the method of mixtures (see [7]), we conclude that

$$
\mathbb{P}\left(\exists n \in \mathbb{N}^*, \, n\,\mathrm{KL}(\widehat{p}_n, p) > (m-1)\log\left(e(1 + n/(m-1))\right) + \log(1/\delta)\right) \le \delta.
$$

$\square$

**Lemma 3.** *For $q, p \in \Sigma_m$ and $\lambda \in \mathbb{R}^{m-1}$,*

$$
\langle \lambda, q \rangle - \varphi_p(\lambda) = \mathrm{KL}(q, p) - \mathrm{KL}(q, p^\lambda),
$$

*where $\varphi_p(\lambda) = \log(p_m + \sum_{k=1}^{m-1} p_k e^{\lambda_k})$ and $p^\lambda = \nabla\varphi_{p_0}(\lambda)$.*

*Proof.* There is a more general way than the ad hoc one below to prove the result. First note that

$$p_k^\lambda = \frac{p_k e^{\lambda_k}}{p_m + \sum_{\ell=1}^{m-1} p_\ell e^{\lambda_\ell}}\,,$$

which implies that

$$p_m + \sum_{k=1}^{m-1} p_k e^{\lambda_k} = \frac{p_m}{p_m^\lambda}, \qquad \lambda_k = \log \frac{p_k^\lambda}{p_k} + \log \frac{p_m}{p_m^\lambda}\,.$$

Therefore we get

$$\begin{aligned}
\langle \lambda, q \rangle - \varphi_p(\lambda) &= \sum_{k=1}^{m-1} q_k \log \left( \frac{p_k^\lambda}{p_k} \frac{p_m}{p_m^\lambda} \right) - \log \left( p_m + \sum_{k=1}^{m-1} p_k e^{\lambda_k} \right) \\
&= \sum_{k=1}^{m-1} q_k \log \frac{p_k^\lambda}{p_k} + (1 - q_m) \log \frac{p_m}{p_m^\lambda} - \log \frac{p_m}{p_m^\lambda} \\
&= \sum_{k=1}^{m} q_k \log \frac{p_k^\lambda}{p_k} = \mathrm{KL}(q, p) - \mathrm{KL}(q, p^\lambda)\,.
\end{aligned}$$

$\square$

## C.2 Deviation Inequality for Bounded Distribution

Let $X_1, X_2, \ldots, X_n, \ldots$ be i.i.d. samples from a distribution $\nu$ of mean $\mu$ supported on $[0, 1]$. We denote by $\widehat{\mu}_n$ the empirical mean

$$\widehat{\mu}_n = \frac{1}{n} \sum_{\ell=1}^{n} X_\ell\,.$$

It is well known, see [11], that we can "project" the distribution $\nu$ on a Bernoulli distribution with the same mean and then use deviation inequality for Bernoulli to concentrate the empirical mean. This method dos not lead to the sharpest confidence intervals but it provides a good trade-off between complexity computation and accuracy.

**Proposition 2.** *For all distribution $\nu$ of mean $\mu$ supported on the unit interval, for all $\delta \in [0, 1]$,*

$$\mathbb{P}\left( \exists n \in \mathbb{N}^*,\, n \, \mathrm{kl}(\widehat{\mu}_n, \mu) > \log(1/\delta) + \log\left( e(1+n) \right) \right) \leq \delta\,.$$

*Proof.* First note that we can upper bound the log-partition function of $\nu$ by the one of a Bernoulli $\mathcal{B}\mathrm{er}(\mu)$, for all $\lambda \in \mathbb{R}$,

$$\log\left( \mathbb{E}[e^{\lambda X_n}] \right) \leq \log\left( \mathbb{E}[X_n e^\lambda + 1 - X_n] \right) = \log\left( 1 - \mu + \mu e^\lambda \right) = \varphi_\mu(\lambda).$$

Then we can follow the proof of Proportion 1 with $m = 2$ and where $M_n^\lambda$ is only a supermartingale but this does not change the result as the property (6) still holds. Thus the proposition follows by specifying Proposition 1 to the case $m = 2$. $\square$

## C.3 Deviation Inequality for sequence of Bernoulli Random Variables

Let $X_1, X_2, \ldots, X_n, \ldots$ be a sequence of Bernoulli random variables adapted to the filtration $(\mathcal{F}_t)_{t \in \mathbb{N}}$. We restate here Lemma F.4. of [6].

**Proposition 3.** *If we denote $p_n = \mathbb{P}(X_n = 1 | \mathcal{F}_{n-1})$, then for all $\delta \in (0, 1]$*

$$\mathbb{P}\left( \exists n \in \mathbb{N}^* : \sum_{\ell=1}^{n} X_\ell < \sum_{\ell=1}^{n} p_\ell/2 - \log(1/\delta) \right) \leq \delta\,.$$

## C.4 Proof of Lemma 2

We just prove that each event forming $\mathcal{E} = \mathcal{E}^r \cap \mathcal{E}^p \cap \mathcal{E}^n$ holds with high probability. For the first one using Proposition 2, since the reward are bounded in the unit interval we have

$$\mathbb{P}\big((\mathcal{E}^r)^c\big) \leq \sum_{h \in [H]} \sum_{(s_h,a_h) \in \mathcal{S} \times \mathcal{A}} \mathbb{P}\big(\exists t \in \mathbb{N}^* : n_h^t(s_h,a_h) \operatorname{kl}\big(\hat{r}_h^t(s_h,a_h), r_h(s_h,a_h)\big) > \beta_r(n_h^t(s_h,a_h),\delta)\big)$$

$$\leq \sum_{h \in [H]} \sum_{(s_h,a_h) \in \mathcal{S} \times \mathcal{A}} \frac{\delta}{3AS^H} \leq \frac{\delta}{3}.$$

where we used Doob's optional skipping in the second inequality in order to apply Proposition 2, see Section 4.1 of [12]. Similarly for the confidence regions for the probabilities transitions, using Proposition 1 we obtain

$$\mathbb{P}\big((\mathcal{E}^p)^c\big) \leq \sum_{h \in [H]} \sum_{(s_h,a_h) \in \mathcal{S} \times \mathcal{A}} \mathbb{P}\big(\exists t \in \mathbb{N}^* : n_h^t(s_h,a_h) \operatorname{KL}\big(\hat{p}_h^t(\cdot|s_h,a_h), p_h(\cdot|s_h,a_h)\big) > \beta_p(n_h^t(s_h,a_h),\delta)\big)$$

$$\leq \sum_{h \in [H]} \sum_{(s_h,a_h) \in \mathcal{S} \times \mathcal{A}} \frac{\delta}{3AS^H} \leq \frac{\delta}{3}.$$

It remains to control the counts, using Proposition 3,

$$\mathbb{P}\big((\mathcal{E}^{\mathrm{cnt}})^c\big) \leq \sum_{h \in [H]} \sum_{(s_h,a_h) \in \mathcal{S} \times \mathcal{A}} \mathbb{P}\left(\exists t \in \mathbb{N}^* : n_h^t(s_h,a_h) < \frac{1}{2}\bar{n}_h^t(s_h,a_h) - \beta^{\mathrm{cnt}}(\delta)\right)$$

$$\leq \sum_{h \in [H]} \sum_{(s_h,a_h) \in \mathcal{S} \times \mathcal{A}} \frac{\delta}{3AS^H} \leq \frac{\delta}{3},$$

where we used that by definition of the pseudo-counts

$$\bar{n}_h^t(s_h,a_h) = \sum_{\ell=1}^t \mathbb{P}\left((s_h^\ell, a_h^\ell) = (s_h,a_h)|\mathcal{F}_{\ell-1}\right),$$

and $\mathcal{F}_{\ell-1}$ is the information available to the agent at step $\ell$. An union bound allows us to conclude

$$\mathbb{P}(\mathcal{E}^c) \leq \mathbb{P}\big((\mathcal{E}^r)^c\big) + \mathbb{P}\big((\mathcal{E}^p)^c\big) + \mathbb{P}\big((\mathcal{E}^{\mathrm{cnt}})^c\big) \leq \delta.$$

# D  Proof of Theorem 1

In this section we present the proof of Theorem 1, which relies on three important ingredients. The first ingredient is Lemma 5 in Appendix D.1, which provides a relationship between the state-action gaps and the diameter $D_h^t(s_h,a_h) := U_h^t(s_h,a_h) - L_h^t(s_h,a_h)$ of the confidence intervals. The second ingredient is Lemma 8 in Appendix D.2, which provides an upper bound on the diameter $D_h^t(s_h,a_h)$. The third ingredient is Lemma 9 in Appendix D.3, which relates the actual counts of state-action pairs to the corresponding pseudo-counts. After providing these ingredients, we present the detailed proof of Theorem 1 in Appendix D.4.

## D.1  Relating state-action gaps to diameters

Before stating Lemma 5, we prove an important property of the UGapE algorithm. We recall that $b^t$ and $c^t$ are the candidate best action and its challenger, defined as

$$b^t = \operatorname*{argmin}_b \left[\max_{a \neq b} U_1^t(s_1,a) - L_1^t(s_1,b)\right],$$

$$c^t = \operatorname*{argmax}_{c \neq b^t} U_1^t(s_1,c).$$

The policy at the root is then defined as $\pi_1^{t+1}(s_1) = \operatorname*{argmax}_{b \in \{b^t, c^t\}} [U_1^t(s_1,b) - L_1^t(s_1,b)]$.

**Lemma 4.** *For all $t \in [\tau_\delta - 1]$, the following inequalities hold:*

1. $U_1^t(s_1, c^t) - L_1^t(s_1, b^t) \le U_1^t(s_1, \pi^{t+1}(s_1)) - L_1^t(s_1, \pi^{t+1}(s_1))$,

2. $U_1^t(s_1, b^t) - L_1^t(s_1, c^t) < 2\left[U_1^t(s_1, \pi^{t+1}(s_1)) - L_1^t(s_1, \pi^{t+1}(s_1))\right]$.

*Proof.* We show the first part by contradiction. If the inequality does not hold, we obtain

$$U_1^t(s_1, b^t) - L_1^t(s_1, b^t) \le U_1^t(s_1, \pi_1^{t+1}(s_1)) - L_1^t(s_1, \pi_1^{t+1}(s_1)) < U_1^t(s_1, c^t) - L_1^t(s_1, b^t),$$
$$U_1^t(s_1, c^t) - L_1^t(s_1, c^t) \le U_1^t(s_1, \pi_1^{t+1}(s_1)) - L_1^t(s_1, \pi_1^{t+1}(s_1)) < U_1^t(s_1, c^t) - L_1^t(s_1, b^t)$$
$$= \max_{a \neq b^t} U_1^t(s_1, a) - L_1^t(s_1, b^t) \le \max_{a \neq c^t} U_1^t(s_1, a) - L_1^t(s_1, c^t),$$

where the last inequality follows from the definition of $b^t$. Combining the two inequalities yields $U_1^t(s_1, b^t) < U_1^t(s_1, c^t) < \max_{a \neq c^t} U_1^t(s_1, a)$, which contradicts the definition of $c^t$.

For the second part, if $t < \tau_\delta$ then the algorithm has not yet stopped, implying

$$U_1^t(s_1, b^t) - L_1^t(s_1, c^t) = U_1^t(s_1, b^t) - L_1^t(s_1, b^t) + U_1^t(s_1, c^t) - L_1^t(s_1, c^t)$$
$$- \left[U_1^t(s_1, c^t) - L_1^t(s_1, b^t)\right]$$
$$< 2\left[U_1^t(s_1, \pi_1^{t+1}(s_1)) - L_1^t(s_1, \pi_1^{t+1}(s_1))\right] - \varepsilon.$$

$\square$

As a consequence of Lemma 4, we can upper bound any confidence interval involving $b^t$ and $c^t$.

**Corollary 3.** *For each pair of actions $a, a' \in \{b^t, c^t\}$, it holds that*

$$U_1^t(s_1, a) - L_1^t(s_1, a') \le 2\left[U_1^t(s_1, \pi^{t+1}(s_1)) - L_1^t(s_1, \pi^{t+1}(s_1))\right].$$

We are now ready to state Lemma 5.

**Lemma 5.** *If $\mathcal{E}$ holds and $t < \tau_\delta$, for all $h \in [H]$ and $s_h \in \mathcal{S}_h(\pi^{t+1})$,*

$$\tilde{\Delta}_h(s_h, \pi_h^{t+1}(s_h)) \le 2\left[U_h^t(s_h, \pi_h^{t+1}(s_h)) - L_h^t(s_h, \pi_h^{t+1}(s_h))\right].$$

*Proof.* The proof for $h \in [2, H]$ is immediate from the correctness of the confidence bounds implied by $\mathcal{E}$, and the fact that the selection is optimistic:

$$\Delta_h(s_h, \pi_h^{t+1}(s_h)) = Q_h(s_h, \pi_h^\star(s_h)) - Q_h(s_h, \pi_h^{t+1}(s_h))$$
$$\le \max_a U_h^t(s_h, a) - L_h^t(s_h, \pi_h^{t+1}(s_h)) = U_h^t(s_h, \pi_h^{t+1}(s_h)) - L_h^t(s_h, \pi_h^{t+1}(s_h)).$$

For $h = 1$, we prove separately that each term in the max is smaller that the right hand side of desired inequality, that is

$$\max\left(\Delta_1(s_1, \pi^{t+1}(s_1)); \Delta; \varepsilon\right) \le 2\left[U_h^t(s_h, \pi_h^{t+1}(s_h)) - L_h^t(s_h, \pi_h^{t+1}(s_h))\right].$$

Now, by definition of the stopping rule, if $t < \tau_\delta$, $U_1^t(s_1, c^t) - L_1^t(s_1, b^t) > \varepsilon$. Using the first property in Lemma 4 yields

$$\varepsilon < U_1^t(s_1, \pi^{t+1}(s_1)) - L_1^t(s_1, \pi^{t+1}(s_1)). \tag{7}$$

Then, exploiting the fact that the action with largest UCB is either $b^t$ or $c^t$, it holds on $\mathcal{E}$ that

$$\Delta_1(s_1, \pi^{t+1}(s_1)) = Q_1(s_1, a^\star) - Q_1(s_1, \pi^{t+1}(s_1))$$
$$\le \max_a U_1^t(s_1, a) - L_1^t(s_1, \pi^{t+1}(s_1))$$
$$= \max_{a \in \{b^t, c^t\}} U_1^t(s_1, a) - L_1^t(s_1, \pi^{t+1}(s_1)).$$

Using Corollary 3 to further upper bound the right hand side yields

$$\Delta_1(s_1, \pi^{t+1}(s_1)) < 2\left[U_1^t(s_1, \pi^{t+1}(s_1)) - L_1^t(s_1, \pi^{t+1}(s_1))\right]. \tag{8}$$

Finally, one can also write, on the event $\mathcal{E}$,

$$
\begin{aligned}
\Delta = \min_{a \neq a^\star} [Q_1(s_1, a^\star) - Q_1(s_1, a)] \ &\leq \ U_1^t(s_1, a^\star) - \max_{a \neq a^\star} Q_1(s_1, a) \\
&\leq \ \max_{a' \in \{b^t, c^t\}} U_1^t(s_1, a') - \min_{a \in \{b^t, c^t\}} Q_1(s_1, a) \\
&\leq \ \max_{a' \in \{b^t, c^t\}} U_1^t(s_1, a') - \min_{a \in \{b^t, c^t\}} L_1^t(s_1, a).
\end{aligned}
$$

In each of the four possible choices of $(a, a')$, Corollary 3 implies that

$$
\Delta \leq 2 \left[ U_1^t \left( s_1, \pi^{t+1}(s_1) \right) - L_1^t \left( s_1, \pi^{t+1}(s_1) \right) \right]. \tag{9}
$$

Lemma 5 follows by combining (7), (8) and (9) with the definition of $\Delta_1^\star \left( s_1, \pi^{t+1}(s_1) \right)$. $\qquad \square$

### D.2 Upper bounding the diameters

In this section we state and prove Lemma 8. We use the notation $\sigma_h = \sum_{i=0}^{h-1} \gamma^i$ to upper bound the discounted reward in $h$ steps. As a first step, we prove the following auxiliary lemma.

**Lemma 6.** *If $\mathcal{E}$ holds, for each $h \in [H]$, each $(s_h, a_h)$ and each $q \in \mathcal{C}_h^t(s_h, a_h)$,*

$$
\sum_{s'} \left( q(s'|s_h, a_h) - p_h(s'|s_h, a_h) \right) U_{h+1}^t(s', \pi_{h+1}^{t+1}(s')) \leq 2\sqrt{2}\sigma_{H-h} \sqrt{\frac{\beta(n_h^t(s_h, a_h), \delta)}{n_h^t(s_h, a_h) \vee 1}}.
$$

*Proof.* First note that for each state $s'$, $U_{h+1}^t(s', \pi_{h+1}^{t+1}(s'))$ can be expressed as an expectation on the form $\mathbb{E}^{\pi^{t+1}} \left\{ \sum_{i=h+1}^{H} \gamma^{i-h-1} u_i^t(s_i, a_i) \mid s_{h+1} = s' \right\}$, which is upper bounded by $\sum_{i=h+1}^{H} \gamma^{i-h-1} = \sigma_{H-h}$ since $u_i^t(s_i, a_i) \leq 1$ for each $(s_i, a_i)$. Note that for $h = H$, $\sigma_{H-H} = \sigma_0 = 0$. If $n_h^t(s_h, a_h) = 0$ the result trivially holds by the conventions adopted for the confidence bounds and regions. Now, if $n_h^t(s_h, a_h) > 0$, we have

$$
\begin{aligned}
\sum_{s'} & \left( q(s'|s_h, a_h) - p_h(s'|s_h, a_h) \right) U_{h+1}^t(s', \pi_{h+1}^{t+1}(s')) \\
&\leq \|q(\cdot|s_h, a_h) - p_h(\cdot|s_h, a_h)\|_1 \, \|U_{h+1}^t(\cdot, \pi_{h+1}^{t+1}(\cdot))\|_\infty \\
&\leq \sigma_{H-h} \left( \|q(\cdot|s_h, a_h) - \hat{p}_h^t(\cdot|s_h, a_h)\|_1 + \|p_h(\cdot|s_h, a_h) - \hat{p}_h^t(\cdot|s_h, a_h)\|_1 \right) \\
&\leq \sigma_{H-h} \left( \sqrt{2\,\mathrm{KL}(\hat{p}_h^t(\cdot|s_h, a_h), q(\cdot|s_h, a_h))} + \sqrt{2\,\mathrm{KL}(\hat{p}_h^t(\cdot|s_h, a_h), p_h(\cdot|s_h, a_h))} \right) \\
&\leq 2\sqrt{2}\sigma_{H-h} \sqrt{\frac{\beta(n_h^t(s_h, a_h), \delta)}{n_h^t(s_h, a_h) \vee 1}},
\end{aligned}
$$

where we have used Pinsker's inequality to bound the $L^1$-norm using the KL divergence, combined with the fact that both $q$ and $p$ are close to the empirical transition probabilities $\hat{p}^t$ under $\mathcal{E}$. $\qquad \square$

As a consequence, we can express the upper bound $U^t$ in terms of the true transition probabilities $p$.

**Corollary 4.** *If $\mathcal{E}$ holds, for each $h \in [H]$ and each $(s_h, a_h)$,*

$$
U_h^t(s_h, a_h) \leq u_h^t(s_h, a_h) + \gamma \sum_{s'} p_h(s'|s_h, a_h) U_{h+1}^t(s', \pi_{h+1}^{t+1}(s')) + 2\sqrt{2}\gamma\sigma_{H-h} \sqrt{\frac{\beta(n_h^t(s_h, a_h), \delta)}{n_h^t(s_h, a_h) \vee 1}}.
$$

We can also express the lower bound $L^t$ in terms of the transition probabilities $p$ and policy $\pi^{t+1}$.

**Lemma 7.** *If $\mathcal{E}$ holds, for each $h \in [H]$ and each $(s_h, a_h)$,*

$$
L_h^t(s_h, a_h) \geq \ell_h^t(s_h, a_h) + \gamma \sum_{s'} p_h(s'|s_h, a_h) L_{h+1}^t(s', \pi_{h+1}^{t+1}(s')) - 2\sqrt{2}\gamma\sigma_{H-h} \sqrt{\frac{\beta(n_h^t(s_h, a_h), \delta)}{n_h^t(s_h, a_h) \vee 1}}.
$$

*Proof.* We exploit the fact that for each $h \in [H]$, each $(s_h, a_h)$ and each $q \in \mathcal{C}_h^t(s_h, a_h)$,

$$\sum_{s'} (q(s'|s_h, a_h) - p_h(s'|s_h, a_h)) \max_{a'} L_{h+1}^t(s', a') \geq -2\sqrt{2}\sigma_{H-h}\sqrt{\frac{\beta(n_h^t(s_h, a_h), \delta)}{n_h^t(s_h, a_h) \vee 1}}.$$

The proof is analogous to the proof of Lemma 6. We can now write

$$L_h^t(s_h, a_h) = \ell_h^t(s_h, a_h) + \gamma \min_{p \in \mathcal{C}_h^t(s_h, a_h)} \sum_{s'} p(s'|s_h, a_h) \max_{a'} L_{h+1}^t(s', a')$$

$$\geq \ell_h^t(s_h, a_h) + \gamma \sum_{s'} p_h(s'|s_h, a_h) \max_{a'} L_{h+1}^t(s', a') - 2\sqrt{2}\gamma\sigma_{H-h}\sqrt{\frac{\beta(n_h^t(s_h, a_h), \delta)}{n_h^t(s_h, a_h) \vee 1}}$$

$$\geq \ell_h^t(s_h, a_h) + \gamma \sum_{s'} p_h(s'|s_h, a_h) L_{h+1}^t(s', \pi_{h+1}^{t+1}(s')) - 2\sqrt{2}\gamma\sigma_{H-h}\sqrt{\frac{\beta(n_h^t(s_h, a_h), \delta)}{n_h^t(s_h, a_h) \vee 1}}.$$

$\square$

We are now ready to state Lemma 8.

**Lemma 8.** *If $\mathcal{E}$ holds, for all $h \in [H]$, $s_h \in \mathcal{S}_h(\pi^{t+1})$ and $a_h$,*

$$D_h^t(s_h, a_h) \leq \sigma_{H-h+1}\left[4\sqrt{2}\sqrt{\frac{\beta(n_h^t(s_h, a_h), \delta)}{n_h^t(s_h, a_h)}} \wedge 1\right] + \gamma \sum_{s'} p_h(s'|s_h, a_h) D_{h+1}^t(s', \pi_{h+1}^{t+1}(s')).$$

*Proof.* The bound on the diameter follows directly from Corollary 4 and Lemma 7:

$$D_h^t(s_h, a_h) = U_h^t(s_h, a_h) - L_h^t(s_h, a_h)$$

$$\leq \left(u_h^t(s_h, a_h) - \ell_h^t(s_h, a_h)\right) + \gamma \sum_{s'} p_h(s'|s_h, a_h) \left(U_{h+1}^t(s', \pi_{h+1}^{t+1}(s')) - L_{h+1}^t(s', \pi_{h+1}^{t+1}(s'))\right)$$

$$+ 4\sqrt{2}\gamma\sigma_{H-h}\sqrt{\frac{\beta(n_h^t(s_h, a_h), \delta)}{n_h^t(s_h, a_h) \vee 1}}$$

$$\leq 4\sqrt{2}\sigma_{H-h+1}\sqrt{\frac{\beta(n_h^t(s_h, a_h), \delta)}{n_h^t(s_h, a_h) \vee 1}} + \gamma \sum_{s'} p_h(s'|s_h, a_h) D_{h+1}^t(s', \pi_{h+1}^{t+1}(s')),$$

where we used $\mathcal{E}^r \supseteq \mathcal{E}$ and Pinsker's inequality to bound

$$u_h^t(s_h, a_h) - \ell_h^t(s_h, a_h) \leq \sqrt{\frac{2\beta(n_h^t(s_h, a_h), \delta)}{n_h^t(s_h, a_h) \vee 1}} < 4\sqrt{2}\sqrt{\frac{\beta(n_h^t(s_h, a_h), \delta)}{n_h^t(s_h, a_h) \vee 1}}.$$

To obtain the final expression in Lemma 8, we observe that it also trivially holds that

$$D_h^t(s_h, a_h) \leq \sigma_{H-h+1} \leq \sigma_{H-h+1} + \gamma \sum_{s'} p_h(s'|s_h, a_h) D_{h+1}^t(s', \pi_{h+1}^{t+1}(s')),$$

hence

$$D_h^t(s_h, a_h) \leq \sigma_{H-h+1} \min\left[4\sqrt{2}\sqrt{\frac{\beta(n_h^t(s_h, a_h), \delta)}{n_h^t(s_h, a_h) \vee 1}}, 1\right] + \gamma \sum_{s'} p_h(s'|s_h, a_h) D_{h+1}^t(s', \pi_{h+1}^{t+1}(s')).$$

The conclusion follows by observing that one can get rid of the maximum with 1 in the denominator by using instead the convention $1/0 = +\infty$. $\square$

### D.3 Relating counts to pseudo-counts

We now assume that the event $\mathcal{E}$ holds and fix some $h \in [H]$ and some state-action pair $(s_h, a_h)$. For every $\ell \geq h$, we define $p_{h,\ell}^\pi(s, a|s_h, a_h)$ to be the probability that starting from $(s_h, a_h)$ in step $h$

and following $\pi$ thereafter, we end up in $(s,a)$ in step $\ell$. We use $p_{h,\ell}^t(s,a|s_h,a_h)$ as a shorthand for $p_{h,\ell}^{\pi^t}(s,a|s_h,a_h)$.

Introducing the *conditional pseudo-counts* $\bar{n}_{h,\ell}^t(s,a;s_h,a_h) := \sum_{i=1}^t p_h^i(s_h,a_h)p_{h,\ell}^i(s,a|s_h,a_h)$ and using that on the event $\mathcal{E}^{\mathrm{cnt}} \supseteq \mathcal{E}$ the counts are close to the pseudo-counts, one can prove:

**Lemma 9.** *If the event $\mathcal{E}^{\mathrm{cnt}}$ holds,* $\left[ \sqrt{\frac{\beta(n_\ell^t(s,a),\delta)}{n_\ell^t(s,a)}} \wedge 1 \right] \leq 2\sqrt{\frac{\beta(\bar{n}_{h,\ell}^t(s,a;s_h,a_h),\delta)}{\bar{n}_{h,\ell}^t(s,a;s_h,a_h)\vee 1}}.$

*Proof.* As the event $\mathcal{E}^{\mathrm{cnt}}$ holds, we know that for all $t < \tau$,

$$
\begin{aligned}
n_\ell^t(s,a) &\geq \frac{1}{2}\bar{n}_\ell^t(s,a) - \beta^{\mathrm{cnt}}(\delta) \\
&\geq \frac{1}{2}\bar{n}_{h,\ell}^t(s,a;s_h,a_h) - \beta^{\mathrm{cnt}}(\delta).
\end{aligned}
$$

We now distinguish two cases. First, if $\beta^{\mathrm{cnt}}(\delta) \leq \frac{1}{4}\bar{n}_{h,\ell}^t(s,a;s_h,a_h)$, then

$$
\sqrt{\frac{\beta(n_\ell^t(s,a),\delta)}{n_\ell^t(s,a)}} \leq \sqrt{\frac{\beta\left(\frac{1}{4}\bar{n}_{h,\ell}^t(s,a;s_h,a_h),\delta\right)}{\frac{1}{4}\bar{n}_{h,\ell}^t(s,a;s_h,a_h)}} \leq 2\sqrt{\frac{\beta\left(\bar{n}_{h,\ell}^t(s,a;s_h,a_h),\delta\right)}{\bar{n}_{h,\ell}^t(s,a;s_h,a_h)\vee 1}},
$$

where we use that $x \mapsto \sqrt{\beta(x,\delta)/x}$ is non-increasing for $x \geq 1$, $x \mapsto \beta(x,\delta)$ is non-decreasing, and $\beta^{\mathrm{cnt}}(\delta) \geq 1$. If $\beta^{\mathrm{cnt}}(\delta) > \frac{1}{4}\bar{n}_{h,\ell}^t(s,a;s_h,a_h)$, simple algebra shows that

$$
1 < 2\sqrt{\frac{\beta^{\mathrm{cnt}}(\delta)}{\bar{n}_{h,\ell}^t(s,a;s_h,a_h)\vee 1}} \leq 2\sqrt{\frac{\beta(\bar{n}_{h,\ell}^t(s,a;s_h,a_h),\delta)}{\bar{n}_{h,\ell}^t(s,a;s_h,a_h)\vee 1}},
$$

where we use that $\beta^{\mathrm{cnt}}(\delta) \leq \beta(0,\delta)$ and $x \mapsto \beta(x,\delta)$ is non-decreasing. If $\bar{n}_{h,\ell}^t(s,a;s_h,a_h) < 1$, the expression uses the trivial bound $\beta^{\mathrm{cnt}}(\delta) > \frac{1}{4}$. In both cases, we have

$$
\left[ \sqrt{\frac{\beta(n_\ell^t(s,a),\delta)}{n_\ell^t(s,a)}} \wedge 1 \right] \leq 2\sqrt{\frac{\beta(\bar{n}_{h,\ell}^t(s,a;s_h,a_h),\delta)}{\bar{n}_{h,\ell}^t(s,a;s_h,a_h)\vee 1}}.
$$

$\square$

## D.4 Detailed proof of Theorem 1

We assume that the event $\mathcal{E}$ holds and fix some $h \in [H]$ and some state-action pair $(s_h,a_h)$. We define some notion of expected diameter in a future step $\ell$ given that $(s_h,a_h)$ is visited at step $h$ under policy $\pi^{t+1}$. For every $(h,\ell) \in [H]^2$ such that $h \leq \ell$ we let

$$
q_{h,\ell}^t(s_h,a_h) := \sum_{(s,a)} p_h^{t+1}(s_h,a_h)p_{h,\ell}^{t+1}(s,a|s_h,a_h)D_\ell^t(s,a).
$$

To be more accurate, $q_{h,\ell}^t(s_h,a_h)$ is equal to the probability that $(s_h,a_h)$ is visited by $\pi^{t+1}$, multiplied by the expected diameter of the state-action pair $(s,a)$ that is reached at step $\ell$ if one applies $\pi^{t+1}$ after choosing $a_h$ in state $s_h$. In particular, $q_{h,\ell}^t(s_h,a_h) = 0$ if $a_h \neq \pi^{t+1}(s_h)$.

**Step 1: lower bounding $q_{h,h}^t(s_h,a_h)$ in terms of the gaps**   From the above definition,

$$
q_{h,h}^t(s_h,a_h) = p_h^{t+1}(s_h,a_h)D_h^t(s_h,a_h).
$$

Using Lemma 5 and the fact that $p_h^{t+1}(s_h,a_h) = 0$ if $a_h \neq \pi^{t+1}(s_h)$ yields

$$
\text{if } t < \tau, \quad q_{h,h}^t(s_h,a_h) \geq \frac{1}{2}p_h^{t+1}(s_h,a_h)\Delta_h(s_h,a_h). \tag{10}
$$

**Step 2: upper bounding $q_{h,h}^t(s_h, a_h)$ in terms of the counts**    Using Lemma 8 and the fact that

$$\sum_{(s,a)} p_h^{t+1}(s_h, a_h) p_{h,\ell}^{t+1}(s, a | s_h, a_h) \left[ \sum_{(s',a')} p_\ell(s'|s,a) \mathbb{1}\left(a' = \pi_{\ell+1}^{t+1}(s')\right) D_{\ell+1}^t(s', a') \right]$$

$$= \sum_{(s',a')} p_h^{t+1}(s_h, a_h) \underbrace{\left[ \sum_{(s,a)} p_{h,\ell}^{t+1}(s, a | s_h, a_h) p_\ell(s'|s,a) \mathbb{1}\left(a' = \pi_{\ell+1}^{t+1}(s')\right) \right]}_{=p_{h,\ell+1}^{t+1}(s',a'|s_h,a_h)} D_{\ell+1}^t(s', a'),$$

one can establish the following relationship between $q_{h,\ell}^t(s_h, a_h)$ and $q_{h,\ell+1}^t(s_h, a_h)$:

$$q_{h,\ell}^t(s_h, a_h) \leq \sum_{(s,a)} p_h^{t+1}(s_h, a_h) p_{h,\ell}^{t+1}(s, a | s_h, a_h) \left[ 4\sqrt{2}\sqrt{\frac{\beta(n_h^t(s,a), \delta)}{n_h^t(s,a)}} \wedge 1 \right] + \gamma q_{h,\ell+1}^{t+1}(s_h, a_h).$$

By induction, one then obtains the following upper bound:

$$q_{h,h}^t(s_h, a_h) \leq \sum_{\ell=h}^{H} \gamma^{\ell-h} \sigma_{H-\ell+1} \sum_{(s,a)} p_h^{t+1}(s_h, a_h) p_{h,\ell}^{t+1}(s, a | s_h, a_h) \left[ 4\sqrt{2}\sqrt{\frac{\beta(n_\ell^t(s,a), \delta)}{n_\ell^t(s,a)}} \wedge 1 \right]. \quad (11)$$

**Step 3: summing the inequalities to get an upper bound on $\overline{n}_h^t(s_h, a_h)$**    Summing for $t \in \{0, \dots, \tau - 1\}$ the inequalities given by (10) yields

$$\sum_{t=0}^{\tau-1} q_{h,h}^t \geq \frac{\tilde{\Delta}_h(s_h, a_h)}{2} \left( \sum_{t=0}^{\tau-1} p_h^{t+1}(s_h, a_h) \right) = \frac{\tilde{\Delta}_h(s_h, a_h)}{2} \overline{n}_h^\tau(s_h, a_h).$$

Summing the upper bounds in (11) yields that $\tilde{\Delta}_h(s_h, a_h) n_h^\tau(s_h, a_h)$ is upper bounded by

$$B_h^\tau(s_h, a_h) := 2 \sum_{t=0}^{\tau-1} \sum_{\ell=h}^{H} \gamma^{\ell-h} \sigma_{H-\ell+1} \sum_{(s,a)} p_h^{t+1}(s_h, a_h) p_{h,\ell}^{t+1}(s, a | s_h, a_h) \left[ 4\sqrt{2}\sqrt{\frac{\beta(n_\ell^t(s,a), \delta)}{n_\ell^t(s,a)}} \wedge 1 \right].$$

The rest of the proof consists in upper bounding $B_h^\tau(s_h, a_h)$ in terms of the pseudo counts $\overline{n}_h^\tau(s_h, a_h)$.

**Step 4: from counts to pseudo-counts**    For all $\ell \geq h$, we introduce the set $\mathcal{S}_\ell(s_h, a_h)$ of states-action pairs $(s,a)$ that can be reached at step $\ell$ from $(s,a)$.

For each $(s,a) \in \mathcal{S}_\ell(s_h, a_h)$, we define

$$C_\ell(s, a; s_h, a_h) = \sum_{t=0}^{\tau-1} p_h^{t+1}(s_h, a_h) p_{h,\ell}^{t+1}(s, a | s_h, a_h) \left[ 4\sqrt{2}\sqrt{\frac{\beta(n_\ell^t(s,a), \delta)}{n_\ell^t(s,a)}} \wedge 1 \right].$$

One can observe that $B_h^\tau(s_h, a_h) = 2 \sum_{\ell=h}^{H} \sum_{(s,a) \in \mathcal{S}_\ell(s_h,a_h)} \gamma^{\ell-h} \sigma_{H-\ell+1} C_\ell(s, a; s_h, a_h)$. To upper bound $C_\ell(s, a; s_h, a_h)$ we further introduce the *conditional pseudo-counts*

$$\overline{n}_{h,\ell}^t(s, a; s_h, a_h) := \sum_{i=1}^{t} p_h^i(s_h, a_h) p_{h,\ell}^i(s, a | s_h, a_h),$$

for which one can write

$$C_\ell(s, a; s_h, a_h) = \sum_{t=0}^{\tau-1} [\overline{n}_{h,\ell}^{t+1}(s, a; s_h, a_h) - \overline{n}_{h,\ell}^t(s, a; s_h, a_h)] \left[ 4\sqrt{2}\sqrt{\frac{\beta(n_\ell^t(s,a), \delta)}{n_\ell^t(s,a)}} \wedge 1 \right].$$

Using Lemma 9 to relate the counts to the conditional pseudo-counts, one can write

$$C_\ell(s, a; s_h, a_h) \leq 8\sqrt{2} \sum_{t=0}^{\tau-1} [\overline{n}_{h,\ell}^{t+1}(s, a; s_h, a_h) - \overline{n}_{h,\ell}^t(s, a; s_h, a_h)] \sqrt{\frac{\beta(\overline{n}_{h,\ell}^t(s, a; s_h, a_h), \delta)}{\overline{n}_{h,\ell}^t(s, a; s_h, a_h) \vee 1}}$$

$$\leq 8\sqrt{2}\sqrt{\beta(\overline{n}_{h,\ell}^\tau(s, a; s_h, a_h), \delta)} \sum_{t=0}^{\tau-1} \frac{\overline{n}_{h,\ell}^{t+1}(s, a; s_h, a_h) - \overline{n}_{h,\ell}^t(s, a; s_h, a_h)}{\sqrt{\overline{n}_{h,\ell}^t(s, a; s_h, a_h) \vee 1}}$$

$$\leq 8\sqrt{2}(1 + \sqrt{2})\sqrt{\beta(\overline{n}_{h,\ell}^\tau(s, a; s_h, a_h), \delta) \times \overline{n}_{h,\ell}^\tau(s, a; s_h, a_h)},$$

where the last step uses Lemma 19 in [17].

Finally, by summing over episodes $\ell$ and over reachable states $(s, a) \in \mathcal{S}_\ell(s_h, a_h)$, we can upper bound $B_h^\tau(s_h, a_h)$ by

$$
2 \sum_{\ell=h}^{H} \gamma^{\ell-h} \sigma_{H-\ell+1} \left[ 8\sqrt{2}(1 + \sqrt{2}) \sqrt{\beta(\overline{n}_h^\tau(s_h, a_h), \delta)} \sum_{(s,a) \in \mathcal{S}_\ell(s_h, a_h)} \sqrt{\overline{n}_{h,\ell}^\tau(s, a; s_h, a_h)} \right]
$$

$$
\leq 2 \sum_{\ell=h}^{H} \gamma^{\ell-h} \sigma_{H-\ell+1} \left[ 8\sqrt{2}(1 + \sqrt{2}) \sqrt{\beta(\overline{n}_h^\tau(s_h, a_h), \delta)} \sqrt{(BK)^{h-\ell}} \sqrt{\sum_{(s,a) \in \mathcal{S}_\ell(s_h, a_h)} \overline{n}_{h,\ell}^\tau(s, a; s_h, a_h)} \right]
$$

$$
= 2 \sum_{\ell=h}^{H} \gamma^{\ell-h} \sigma_{H-\ell+1} \left[ 8\sqrt{2}(1 + \sqrt{2}) \sqrt{\beta(\overline{n}_h^\tau(s_h, a_h), \delta)} \sqrt{(BK)^{h-\ell}} \sqrt{\overline{n}_h^\tau(s_h, a_h)} \right],
$$

where we have used that $\sum_{(s,a) \in \mathcal{S}_\ell(s_h, a_h)} \overline{n}_{h,\ell}^\tau(s, a; s_h, a_h) = \overline{n}_h^\tau(s_h, a_h)$. By using further Lemma 10 to upper bound all the constants, we obtain

$$
B_h^\tau(s_h, a_h) \leq 64\sqrt{2}(1 + \sqrt{2}) \left( \sqrt{BK} \right)^{H-h} \sqrt{\overline{n}_h^\tau(s_h, a_h) \beta(\overline{n}_h^\tau(s_h, a_h), \delta)} .
$$

**Lemma 10.** *For every $x > 1$, $\sum_{\ell=h}^{H} (\gamma x)^{\ell-h} \sigma_{H-\ell+1} \leq \frac{x^{H-h}}{\left(1 - \frac{1}{x}\right)^2}$.*

*Proof.* Since $\gamma \leq 1$ and $x > 1$, we can write

$$
\sum_{\ell=h}^{H} (\gamma x)^{\ell-h} \sigma_{H-\ell+1} \leq \sum_{\ell=h}^{H} x^{\ell-h}(H - \ell + 1) = \sum_{\ell=0}^{H-h} x^\ell(H - h - \ell + 1)
$$

$$
= x^{H-h} \sum_{\ell=0}^{H-h} \frac{H - h - \ell + 1}{x^{H-h-\ell}} = x^{H-h} \sum_{\ell=0}^{H-h} (\ell + 1) r^\ell,
$$

where $r = 1/x < 1$. The latter is an *arithmetico-geometric sum* that can be upper bounded as

$$
\sum_{\ell=0}^{H-h} (\ell + 1) r^\ell \leq \sum_{\ell=0}^{\infty} (\ell + 1) r^\ell = \frac{1}{(1 - r)^2} = \frac{1}{\left(1 - \frac{1}{x}\right)^2} .
$$

$\square$

# E   Proof of Theorem 2

The proof of Theorem 2 uses the same ingredients as the proof of Theorem 1: Lemma 5 which relates the gaps to the diameters of the confidence intervals $D_h^t(s_h, a_h) = U_h^t(s_h, a_h) - L_h^t(s_h, a_h)$ and a counterpart of Lemma 8 for the deterministic case, stated below.

**Lemma 11.** *If $\mathcal{E}$ holds, and $t_{1:H} = (s_1, a_1, \ldots, s_H, a_H)$ is the $(t + 1)$-st trajectory generated by MDP-GapE, for all $h \in [H]$,*

$$
D_h^t(s_h, a_h) \leq \left[ \sqrt{\frac{2\beta(n_h^t(s_h, a_h), \delta)}{n_h^t(s_h, a_h)}} \wedge 1 \right] + \gamma D_{h+1}^t(s_{h+1}, a_{h+1}).
$$

It follows from Lemma 11 that for all $h \in [H]$, along the $(t + 1)$-st trajectory $t_{1:H} = (s_1, a_1, \ldots, s_H, a_H)$,

$$
D_h^t(s_h, a_h) \leq \sum_{\ell=h}^{H} \gamma^{\ell-h} \left[ \sqrt{\frac{2\beta(n_\ell^t(s_\ell, a_\ell), \delta)}{n_\ell^t(s_\ell, a_\ell)}} \wedge 1 \right].
$$

Letting $n^t(t_{1:H})$ be the number of times the trajectory $t_{1:H}$ has been selected by MDP-GapE in the first $t$ episodes, one has $n_\ell^t(s_\ell, a_\ell) \geq n^t(t_{1:H})$. Hence, if $n^t(t_{1:H}) > 0$, it holds that

$$
D_h^t(s_h, a_h) \leq \sum_{\ell=h}^{H} \gamma^{\ell-h} \sqrt{\frac{2\beta(n^t(t_{1:H}), \delta)}{n^t(t_{1:H})}} = \sigma_{H-h+1} \sqrt{\frac{2\beta(n^t(t_{1:H}), \delta)}{n^t(t_{1:H})}}.
$$

Using Lemma 5, if $t < \tau$, if $t_{1:H}$ is the trajectory selected at time $(t+1)$, either $n^t(t_{1:H}) = 0$ or

$$\forall h \in [H], \ \tilde{\Delta}_h(s_h, a_h) \leq \sigma_{H-h+1}\sqrt{\frac{2\beta(n^t(t_{1:H}), \delta)}{n^t(t_{1:H})}}$$

It follows that for any trajectory $t_{1:H}$,

$$n^\tau(t_{1:H})\left[\max_{h \in [H]} \frac{\left(\tilde{\Delta}_h(s_h, a_h)\right)^2}{(\sigma_{H-h+1})^2}\right] \leq 2\beta(n^\tau(t_{1:H}), \delta).$$

The conclusion follows from Lemma 12 and from the fact that $\tau = \sum_{t_{1:H} \in \mathcal{T}} n^\tau(t_{1:H})$.

## F   Sample complexity of Sparse Sampling in the Fixed-Confidence Setting

In this section, we prove Lemma 1.

For simplicity, and without loss of generality, assume that the reward function is known. Let $C > 0$. Sparse Sampling builds, recursively, the estimates $\widehat{V}_h$ and $\widehat{Q}_h$ for $h \in [H+1]$, starting from $\widehat{V}_{H+1}(s) = 0$ and $\widehat{Q}_{H+1}(s, a) = 0$ for all $(s, a)$. Then, from a target state-action pair $(s, a)$, it samples $C$ transitions $Z_i \sim p_h(\cdot|s, a)$ for $i \in [C]$ and computes:

$$\widehat{Q}_h(s, a) = r_h(s, a) + \frac{1}{C}\sum_{i=1}^C \widehat{V}_{h+1}(Z_i), \ \ \text{with} \ \widehat{V}_h(s) = \max_a \widehat{Q}_h(s, a)$$

For an initial state $s$, its output is $\widehat{Q}_1(s, a)$ for all $a \in [K]$. For any state $s$, consider the events

$$\mathcal{G}(s, a, h) = \left\{\left|\widehat{Q}_h(s, a) - Q_h^\star(s, a)\right| \leq \varepsilon_h\right\} \bigcap \left\{\bigcap_{z \in \text{supp}[p_h(\cdot|s,a)]} \mathcal{G}(z, h+1)\right\}.$$

and

$$\mathcal{G}(s, h) = \bigcap_{a \in [K]} \mathcal{G}(s, a, h).$$

defined for $h \in [H+1]$, where $\varepsilon_h := (H - h + 1)H\sqrt{(2/C)\log(2/\delta')}$ for some $\delta' > 0$.

Let

$$\delta_h = \frac{2K\delta'}{BK - 1}\left((BK)^{H-h+1} - 1\right)$$

We prove that, for all $s$ and all $h$, $\mathbb{P}[\mathcal{G}(s, h)] \geq 1 - \delta_h$. We proceed by induction on $h$. For $h = H + 1$, we have $\widehat{Q}_{H+1}(s, a) = Q_{H+1}^\star(s, a) = 0$ for all $(s, a)$ by definition, which gives us $\mathbb{P}[\mathcal{G}(s, a, H+1)] = 1$ and, consequently, $\mathbb{P}[\mathcal{G}(s, H+1)] = 1$.

Now, assume that $\mathbb{P}[\mathcal{G}(z, h+1)] \geq 1 - \delta_h$ for all $z$. Since

$$\left|\widehat{Q}_h(s, a) - Q_h^\star(s, a)\right| \leq \frac{1}{C}\left|\sum_{i=1}^C \left(\widehat{V}_{h+1}(Z_i) - V_{h+1}^\star(Z_i)\right)\right| + \frac{1}{C}\left|\sum_{i=1}^C \left(V_{h+1}^\star(Z_i) - \mathbb{E}\left[V_{h+1}^\star(Z_i)\right]\right)\right|$$

We have,

$$\mathbb{P}\left[\mathcal{G}(s, a, h)^{\complement}\right] \leq \sum_{z \in \text{supp}[p_h(\cdot|s,a)]} \mathbb{P}\left[\mathcal{G}(z, h+1)^{\complement}\right] + \mathbb{P}\left[\frac{1}{C}\left|\sum_{i=1}^C \left(V_{h+1}^\star(Z_i) - \mathbb{E}\left[V_{h+1}^\star(Z_i)\right]\right)\right| \geq \varepsilon_h - \varepsilon_{h+1}\right]$$

$$\leq B\delta_{h+1} + 2\exp\left(-\frac{C(\varepsilon_h - \varepsilon_{h+1})^2}{2H^2}\right) \leq B\delta_{h+1} + 2\delta'$$

and, consequently,

$$\mathbb{P}\left[\mathcal{G}(s,h)^{\complement}\right] \leq BK\delta_{h+1} + 2K\delta' = \delta_h.$$

which gives us $\mathbb{P}\left[\mathcal{G}(s,h)\right] \geq 1 - \delta_h$, as claimed above. In particular, taking $h = 1$, we have

$$\left|\widehat{Q}_1(s,a) - Q_1^\star(s,a)\right| \leq H^2\sqrt{(2/C)\log(2/\delta')}$$

with probability at least $1 - \delta$, where $\delta = 2K\delta'\left((BK)^H - 1\right)/(BK - 1)$. Finally, we let $\varepsilon :=$ $H^2\sqrt{(2/C)\log(2/\delta')}/2$ and solve for $C$, obtaining

$$C = \mathcal{O}\left(\frac{H^5}{\varepsilon^2}\log\left(\frac{BK}{\delta}\right)\right).$$

Thus predicting $\hat{a} = \operatorname*{argmax}_a \widehat{Q}_1(s_1,a)$ after $\mathcal{O}\left(C(BK)^H\right)$ sampled transitions we have

$$\mathbb{P}\left(Q^\star(s_1, \hat{a}_\tau) > Q^\star(s_1, a^\star) - \varepsilon\right) \geq 1 - \delta.$$

# G   A Technical Lemma

We state and prove below a technical result that permits to obtain an upper bound on $n$ from a condition of the form $n\Delta^2 \leq \beta(n, \delta)$, like the one which appears in Theorem 1.

**Lemma 12.** *Let $n \geq 1$ and $a, b, c, d > 0$. If $n\Delta^2 \leq a + b\log(c + dn)$ then*

$$n \leq \frac{1}{\Delta^2}\left[a + b\log\left(c + \frac{d}{\Delta^4}(a + b(\sqrt{c} + \sqrt{d}))^2\right)\right].$$

*Proof.* Since $\log(x) \leq \sqrt{x}$ and $\sqrt{x+y} \leq \sqrt{x} + \sqrt{y}$ for all $x, y > 0$, we have

$$n\Delta^2 \leq a + b\sqrt{c + dn} \leq a + b\sqrt{c} + b\sqrt{d}\sqrt{n}$$

$$\implies \sqrt{n}\Delta^2 \leq \frac{a + b\sqrt{c}}{\sqrt{n}} + b\sqrt{d} \leq a + b(\sqrt{c} + \sqrt{d})$$

$$\implies n \leq \frac{1}{\Delta^4}\left(a + b(\sqrt{c} + \sqrt{d})\right)^2.$$

Hence,

$$n\Delta^2 \leq a + b\log(c + dn)$$

$$\implies n\Delta^2 \leq a + b\log(c + dn) \quad \text{and} \quad n \leq \frac{1}{\Delta^4}\left(a + b(\sqrt{c} + \sqrt{d})\right)^2$$

$$\implies n\Delta^2 \leq a + b\log\left(c + \frac{d}{\Delta^4}\left(a + b(\sqrt{c} + \sqrt{d})\right)^2\right).$$

$\square$