[Reviews · NeurIPS 2020]

Review 1

Summary and Contributions: The paper presents an adaptive MCTS algorithm called MDP-GapE, and proves a gap-dependent sample complexity upper bound. It is also verified empirically that MDP-GapE algorithm is more sample efficient than baseline algorithms.

Strengths: The paper proves a sample complexity upper bound that depends on suboptimal gaps, which improves previous results. Empirically, the sample complexity of MDP-GapE algorithm is an order of magnitude smaller than baseline algorithms.

Weaknesses: The empirical comparison is not totally fair (see comments below).

Correctness: Proofs are morally correct, but the empirical comparison needs more discussion.

Clarity: The paper is self-contained, and the correctness can be verified given modest amount of time. It's better to have more high-level explanation of the algorithm.

Relation to Prior Work: The paper discussed previous results very clearly.

Reproducibility: Yes

Additional Feedback: =====Post-rebuttal===== The authors addressed some of my concerns. As the authors would redesign some of the experiments in the revision, I'd raise my score to 6. Comments and questions: 1. Are there any lower bound results on the sample complexity of planning? 2. In the MDP-GapE algorithm, the rule for selecting actions in the first step is much more complicated than the remaining steps. Are there any particular reasons, and what is the high-level idea of this algorithm? If I understand correctly this rule is to get the gap-dependent sample complexity. What if we use the simple greedy policy for the first action, and what will go wrong in the proof? It's a little bit confusing because algorithm with gap-dependent sample complexity in RL still uses greedy policies [1]. 3. In the experiment, the actual number of samples is much lower than the sample complexity bound, and I'm wondering what is the reason. - In the experiment setting, the exploration function beta is much smaller than that in the theoretical analysis. Therefore the actual number of samples needed will be smaller than the sample complexity upper bound. However, the baseline algorithm (SS) is measured by its theoretical sample complexity upper bound. Although Sparse Sampling is not an adaptive algorithm, it is still possible to reduce the number of samples required by tuning parameters (or exploration functions) of the algorithm. Is it more reasonable to compare the performance with a tuned version of SS? - The actual number of samples is even much smaller than (BK)^{H-1}. In the MCTS world, it means that there are unexplored nodes in the search tree. But in the experiment, BK^{H-1}>>|SA|. So the analysis, in this case, is very loose. Actually I think similar techniques in RL literature suggest that the \sqrt{BK}^{H-1} term in Theorem 1 could be improved to \sqrt{SA}. Is it still fair to test the empirical performance in this setting? Note that the baseline performance of SS is given by the sample complexity *upper bound*, analyzed in general MDPs (e.g., MDPs with unbounded state space). 4. The gap-dependent sample complexity is indeed an improvement. [1] has shown that for RL problems the upper bound can dependent on the suboptimal gap of all state-action pairs, instead of gaps of the first action. Is it possible to get similar results in the planning setting? [1] Max Simchowitz and Kevin G Jamieson. Non-Asymptotic Gap-Dependent Regret Bounds for Tabular MDPs.


Review 2

Summary and Contributions: This paper considers a trajectory-based Monte-Carlo tree search algorithm for planning in MDPs. The algorithm follows the idea of UGapE-MCTS: it uses a best arm identification algorithm to choose the first action in the trajectory, and performs optimistic planning thereafter, by constructing upper and lower confidence bounds on the Q-values. The authors establish a problem-dependent sample complexity of the proposed algorithm for general MDPs with stochastic rewards and transitions. The gap-dependent bounds are interesting and of importance for understanding hard problem instances.

Strengths: Compared to piror work on Monte-Carlo planning algorithm with theoretical guarantees, the proposed algorithm MDP-GapE is shown to achieve a better gap-dependent sample compelxity. Also, MDP-GapE is effective in practice, as shown in the experiments.

Weaknesses: The idea of combing a best-arm-identification and optimistic planning is not new. It has been explored in prior work UGapE-MCTS for two-player games. The proposed MDP-GapE is adapted from UGapE-MCTS, with a different construction of upper/lower confidence bounds. The analysis of these two also appear similar. It would be good if the authors comment on the motivation/advantage of using KL confidence sets. Will similar results hold if using standard confidence bounds as UGapE-MCTS? There is no lower bound, so that it is difficult to state whether the upper bound presented in Corollary 1 is tight.

Correctness: The theory is sound and the proofs seem to be valid (though I did not verify all the details in appendix).

Clarity: Overall, the paper is well written and relatively easy to follow.

Relation to Prior Work: Yes. The following two papers are also related: The finite sample complexity of UCT has recently been investigated in [1]. And another related work establishes the convergence rate of MCTS with entropy regularization [2]. [1] D. Shah, Q. Xie, and Z. Xu. Non-Asymptotic Analysis of Monte Carlo Tree Search, 2019 [2] C. Xiao, R. Huang, J. Mei, D. Schuurmans, and M. Müller. Maximum entropy monte-carlo planning. In Advances in Neural Information Processing Systems, 2019.

Reproducibility: Yes

Additional Feedback: Scaling in epsilon: As pointed out in the paper, the dependence on epsilon is worse compared to previous work. It would be good if the authors comment on where this loss of performance occurs, either due to analysis or the algorithm itself. The authors are suggested to add discussion of the results for benign/hard problem settings to illustrate an improved/worst-case sample complexity. MDP-GapE only adopts BAI for the first step. Will applying BAI algorithm for multiple steps help improve the sample complexity? ===Post-rebuttal update=== The authors addressed most of my questions. Overall the results are interesting and solid. The paper points to some interesting future directions. I would keep my score and suggest accepting this paper.


Review 3

Summary and Contributions: This paper considers the problem of on-line planning in MDPs. Given a state, the problem is to compute an epsilon-optimal action with probability at least 1 - delta. The agent has access to a sampling model that can generate a next state and a reward for any given state and action--the compute time can be taken as the number of calls to this sampling model. The paper presents a sampling algorithm, its analysis (shown to enjoy a tighter bound than previous approaches), and also experimental validation of its efficiency. Addendum: I have read the authors' response. I retain my assessment and score.

Strengths: The paper is very well-written: clear and easy to follow. The algorithm, theoretical analysis, and experimental comparisons are all useful contributions--together making up a good paper.

Weaknesses: No significant weaknesses.

Correctness: Yes.

Clarity: Yes.

Relation to Prior Work: Yes.

Reproducibility: Yes

Additional Feedback: Although you fix a finite horizon H, there appears to be no need to suffix the MDP's rewards r and transitions p by h in [1, 2, , H]--they are h-indepdent, are they not? Please explain the intuition behind having different rules for selecting the first action (greedy w.r.t. a confidence width) and subsequent actions (greedy w.r.t. a UCB). A related question is why you need to go along the generated trajectory. You are allowed to sample any arbitrary state-action pair at each step--so why the choice of sticking to a trajectory starting at s_{1}? In Equation (4), "<= epsilon" must be placed outside the curly brace.

[Author Response · NeurIPS 2020]

We thank the reviewers for their insightful comments and questions, which we answer below.

**(R1) High-level explanation**  The high-level intuition behind MDP-GapE is that unlike a purely optimistic policy,
we do not aim at minimizing regret while learning the best action to take. MDP-GapE indeed explores much more
at depth 1 compared to a UCRL type algorithm. The UGapE rule used at depth 1 is crucial for quickly achieving the
stopping condition that we propose: stop when one of the confidence intervals on the value at depth 1 is larger than and
separated from the others. We will make sure to improve the high-level explanation of the algorithm in our revision.
Note that using a greedy (optimistic) policy at depth 1 would require a different stopping rule, inspired perhaps by
lil'UCB [Jamieson et al. 2013], and a completely different analysis. Hence we did not explore this path.

**(R1,R2) Lower bounds**  The only lower bound on the sample complexity of MCTS planning that we are aware of
it the $(1/\varepsilon)^{1/\log(1/\gamma)}$ worse-case bound of [Kearns et al. 02]. As for problem-dependent lower bounds, there exists
some for $H = 1$, which corresponds to $\varepsilon$-best-arm identification in a multi-armed bandit. In that case, the lower bound
of [Mannor and Tsitsiklis 04] indeed scales with the gaps in step 1. We will add a paragraph on lower bounds in our
revision, in which we will leave the design of problem-dependent lower bounds for $H \geq 2$ as an important future work.

**(R1) Experiments**  It is true that in experiments, we use tighter threshold functions than those prescribed by theory.
However, we did not *tune* these functions to perform well on the studied problems. Their choice is rather inspired by our
theoretical results (for their scaling in $n_h^t(s, a)$), un-doing a few union bounds that were found to be conservative. Albeit
questionable, using threshold functions "slightly smaller than theory" mimics what is sometimes done in the bandit
literature, and is also quite common for UCT users. Regarding Sparse Sampling (SS), we will remove $n_{SS}$ from Table
4 and propose instead the following discussion. The sample complexity of SS with parameter $C$ (number of calls to the
generative model in each node) is $(K^{H+1} - K)/(K - 1)$ for $C = 1$ and of order $\sum_{h=0}^{H-1}[(KC) \times (K(\min(B, C)))^h]$
for larger values of $C$. Thus, beyond very small $C$, the runtime of SS is prohibitively too large to try the algorithm in
our setting (larger than $10^H$). For $C = 1$, the sample complexity of SS is $2.0 \times 10^4$, $4.9 \times 10^5$ and $1.2 \times 10^7$ in the 3
experiments in Table 4, which is larger than the maximal sample complexity observed for MDP-GapE. Still, SS has a
larger simple regret (e.g. we observed $\max_n r_n = 0.43$ for $\varepsilon = 0.5$, $H = 8$).

**(R1) Size of search tree**  We believe that we can indeed replace $(BK)^{H-1}$ with $SA$ in the bound. Yet planning
algorithms are usually intended for the case $(BK)^{H-1} \ll SA$, this is why we focus on the scaling in $(BK)^{H-1}$.
Although this does not hold in our experiments, none of the algorithms that we implemented exploits the knowledge of
$S$ (only that of the support $B$) and we propose to perform experiments with much larger state spaces in the revision.

**(R1) Gaps of all state-action pairs**  For deterministic transitions, Theorem 2 actually provides an upper bound that
involves the gaps in all (reachable) state-action pairs. Yet, beyond this case, we did not manage to get a tight bound of
this flavor. We remark that the bound of Simchowitz and Jamieson (which is on the regret, and is therefore hard to
compare to our sample complexity bound), features a sum over all inverse gaps but also includes a term that is inversely
proportional to the minimum gap among *all* state-action pairs, which can be arbitrarily smaller than the depth 1 gaps.

**(R2) Novelties in MDP-GapE**  The analysis of MDP-GapE is much more sophisticated than that of UGapE-MCTS.
The crucial difference is that for stochastic transitions we need to construct confidence intervals on the expected values,
whose diameter is harder to relate to the number of visits in the tree and required the introduction of *pseudo-counts*. The
proof of Theorem 1 relies on a completely new proof technique that sums the local confidence bounds across episodes.

**(R2) KL confidence sets**  One could replace the KL confidence sets with the L1 confidence sets typically used in
UCRL, and obtain the same guarantees (our current analysis uses Pinsker's inequality and does not fully exploit the KL
confidence sets). Yet as the KL confidence sets are tighter, their practical use leads to earlier stopping.

**(R2) Dependence on $\varepsilon$**  Improving the dependence on $\varepsilon$ is mentioned as a possible future work in our conclusion.

**(R2) Benign/hard planning problems**  Our best intuition is that "hiding" the reward very deep in the tree will result
in hard problems, while providing intermediate reward along the optimal path will cause the algorithm to stop more
quickly. In our revision, we will try to explicit the scaling of our bounds in these two cases.

**(R2) UGapE in every step?**  Our analysis crucially depends on being optimistic at depth $> 1$, and does not permit to
analyze an algorithm using UGapE in every step. Our intuition is that this approach would not be more efficient for
finding the best action at depth 1, although it might be beneficial for finding a good policy for the next steps as well.

**(R3) Depth-dependent dynamics**  A standard MDP would indeed have the same dynamics at every depth., i.e. the
transition probabilities and rewards are depth-independent. However, since most planning algorithms explore the search
tree up to a fixed horizon, they can also handle depth-dependent transition probabilities and rewards without incurring
additional computational complexity, so we include them in order to make the algorithm as general as possible.

**(R3) Trajectory-based planning**  As explained in the introduction, we do not assume that we have access to an arbitrary
generative model, but only to a *forward model* that can sample actions *in the current state*, starting from the root state.

[Meta-Review · NeurIPS 2020]

The reviewers unanimously agree that the paper makes a nice contribution to the MCTS literature and offers novel algorithmic and analytical ideas.